# Single-cell sequencing highlights heterogeneity and malignant progression in actinic keratosis and cutaneous squamous cell carcinoma

Dan-Dan Zou[1,2†], Ya-Zhou Sun[3,4†], Xin-Jie Li[4], Wen-Juan Wu[1], Dan Xu[1], Yu-Tong He[4], Jue Qi[1], Ying Tu[1], Yang Tang[1], Yun-Hua Tu[1], Xiao-Li Wang[5], Xing Li[6], Feng-Yan Lu[7], Ling Huang[8], Heng Long[9], Li He[1]*, Xin Li[4,10]*

[1]Department of Dermatology, First Affiliated Hospital of Kunming Medical University, Yunnan, China; [2]Department of Dermatology, The Affiliated Hospital of Kunming University of Science and Technology, The First People's Hospital of Yunnan Province, Kunming, Yunnan, China; [3]Clinical Big Data Research Center, The Seventh Affiliated Hospital of Sun Yat-sen University, Shenzhen, Guangdong, China; [4]School of Medical, Shenzhen Campus of Sun Yat-sen University, Shenzhen, Guangdong, China; [5]Department of Dermatology, Changzheng Hospital, Naval Medical University, Shanghai, China; [6]Department of Dermatology, People's Hospital of Chuxiong Yi Autonomous Prefecture, Chuxiong, Yunnan, China; [7]Department of Dermatology, Qujing Affiliated Hospital of Kunming Medical University, The First People's Hospital of Qujing, Yunnan, China; [8]Department of Dermatology, First Affiliated Hospital of Dali University, Dali, Yunnan, China; [9]Wenshan Zhuang and Miao Autonomous Prefecture Dermatology Clinic, Wenshan Zhuang and Miao Autonomous Prefecture Specialist Hospital of Dermatology, Wenshan, Yunnan, China; [10]Guangdong Provincial Key Laboratory of Digestive Cancer Research, The Seventh Affiliated Hospital of Sun Yat-sen University, Guangdong, China

**\*For correspondence:**
drheli2662@126.com (LH);
lixin253@mail.sysu.edu.cn (XL)

[†]These authors contributed equally to this work

**Competing interest:** The authors declare that no competing interests exist.

**Abstract** Cutaneous squamous cell carcinoma (cSCC) is the second most frequent of the keratinocyte-derived malignancies with actinic keratosis (AK) as a precancerous lesion. To comprehensively delineate the underlying mechanisms for the whole progression from normal skin to AK to invasive cSCC, we performed single-cell RNA sequencing (scRNA-seq) to acquire the transcriptomes of 138,982 cells from 13 samples of six patients including AK, squamous cell carcinoma in situ (SCCIS), cSCC, and their matched normal tissues, covering comprehensive clinical courses of cSCC. We identified diverse cell types, including important subtypes with different gene expression profiles and functions in major keratinocytes. In SCCIS, we discovered the malignant subtypes of basal cells with differential proliferative and migration potential. Differentially expressed genes (DEGs) analysis screened out multiple key driver genes including transcription factors along AK to cSCC progression. Immunohistochemistry (IHC)/immunofluorescence (IF) experiments and single-cell ATAC sequencing (scATAC-seq) data verified the expression changes of these genes. The functional experiments confirmed the important roles of these genes in regulating cell proliferation, apoptosis, migration, and invasion in cSCC tumor. Furthermore, we comprehensively described the tumor microenvironment (TME) landscape and potential keratinocyte-TME crosstalk in cSCC providing theoretical basis for immunotherapy. Together, our findings provide a valuable resource for deciphering the progression from AK to cSCC and identifying potential targets for anticancer treatment of cSCC.

## Editor's evaluation

This important study delineates the molecular changes driving the progression from actinic keratosis (AK) to cutaneous squamous cell carcinoma (cSCC). Using state-of-the-art single-cell RNA profiling of 138,982 cells from 13 samples of six patients including AK, squamous cell carcinoma in situ (SCCIS), cSCC, and their matched normal tissues, thus covering comprehensive clinical courses of cSCC, the authors provide an invaluable data resource. This study identified several previously unreported and interesting candidate genes involved in different stages of the malignant progression of skin neoplasias, which have been validated in situ, and partially in vitro. These findings substantially advance our understanding of the molecular changes leading to skin cancer.

## Introduction

Invasive cutaneous squamous cell carcinoma (cSCC) is the second most common skin malignancy accounting for 20% of keratinocyte carcinomas and the fatality rate is also second to melanoma (**Stratigos et al., 2020**). The morbidity of cSCC is steadily increasing, posing a significant threat to public health. The most important cause of cSCC is ultraviolet (UV) irradiation from sunlight (**Marks et al., 1988**). The occurrence of UV-induced cSCC is a multi-stage process, and its progression is usually slow (**Boukamp, 2005**). Early detection, diagnosis, and treatment are very important for patients with cSCC in the progressive multi-step process. The most significant risk factor for cSCC is actinic keratosis (AK), a precancerous lesion developed from the damage effects of chronical UV irradiation, which has an extremely high incidence in the elderly. Up to 65–97% of cSCCs are reported to originate in lesions previously diagnosed as AKs (**Röwert-Huber et al., 2007**). The two diseases have a lot in common in terms of etiology, pathogenesis, and genetic characteristics (**Chitsazzadeh et al., 2016**). However, it is difficult to predict whether early precancerous lesions will further develop into invasive tumors (**Schmitz et al., 2018**). Even though significant mutations of important genes closely related to cSCC were also detected in AK (**Inman et al., 2018**), most AK with these mutations did not transform into cSCC. Therefore, there is an urgent research need to define the critical molecular biomarkers and origin cancerous cells driving AK progress to cSCC, which will contribute to the prevention, early diagnosis, and effective treatment of cSCC.

At the same time, the occurrence, development, invasion, and metastasis of tumors are closely related to the tumor microenvironment (TME) (**Parker et al., 2021**). The TME includes immune and inflammatory cells, fibroblasts, microvessels, and biomolecules infiltrated therein around tumor cells (**da Cunha et al., 2019**). During the growth process, tumor cells interact with these cells and extracellular stroma, forming a special TME, affecting the production of chemokines, growth factors, and proteolytic enzymes, and promoting tumor proliferation, angiogenesis, invasion, and metastasis (**Nissinen et al., 2016**). Recently, numerous studies have showed complex cellular communication network between tumor cells and TME in many types of cancer including cSCC (**Bauer et al., 2018**). Thus, the analysis of cell-cell communication in TME of cSCC will help us to understand the potential mechanisms during the progression from AK to cSCC in depth and develop new immunotherapy strategy for cSCC.

However, due to the complex tumor heterogeneity and high mutation load of cSCC (**Sordillo et al., 2018**), it is more difficult to identify the driving genes for the occurrence and development of cSCC. Although a number of cSCC-related genes have been identified, the results in different studies vary greatly (**Inman et al., 2018**; **Thomson et al., 2021**). In addition, due to limitations of technologies, the previous results based on bulk sequencing generally include a mixture of various cells, which may cover up key characteristic changes in tumor development (**Gupta and Kuznicki, 2020**). Single-cell RNA sequencing (scRNA-seq) technology provides a powerful tool for obtaining transcriptome characteristics at the single-cell resolution level. It can not only reveal the heterogeneity of tumor cells and the progress of tumor development, but also reveal the intercellular communication between tumor cells and their TME (**Shao et al., 2020**). Recently, single-cell sequencing technology has been applied in the studies of skin diseases, including skin aging, psoriasis, and cSCC (**Zou et al., 2021**; **Ji et al., 2020**; **Cheng et al., 2018**). However, characterization of the initiation and progression of cSCC, especially the key transformation from AK to cSCC, is still lacking.

In this study, we used scRNA-seq technology to analyze 138,982 cells from 13 samples of six patients including AK, squamous cell carcinoma in situ (SCCIS), cSCC, and their matched normal

tissues, covering comprehensive clinical courses of cSCC and filling the current blank of single-cell profiling of these diseases. Using this unique resource, we identified key cell subpopulations that may play an important role in the development from AK to cSCC. Importantly, we identified the early malignant cell subpopulation in SCCIS and comprehensively analyzed the characteristics related to the malignant status of these cells. Based on the identification of key cell subpopulations, we screened out key candidate genes of each important step in the transformation from normal skin to cSCC. The functional experiment verified that these key genes may play important driving roles in tumorigenesis. In addition, we described the TME landscape and cell-cell crosstalk of poorly differentiated cSCC in details and identified important signaling pathways related to tumor progression. Together, our comprehensive analysis deeply revealed the whole malignant progression from normal skin to cSCC, and uncovered the heterogeneity of cSCC tumors, providing insights into understanding of cSCC initiation and progression and new therapeutic treatment development.

## Results

### Single-cell transcriptome profiling identified different subgroups of keratinocytes in human skin

We generated scRNA-seq profiles of 13 samples from six patients presenting for surgical resection using the 10x Genomics Chromium platform. All these samples included three AK samples, one SCCIS tumor sample, three cSCC tumor samples (low-risk and high-risk) without any treatment, and patient-matched six normal skin samples, which almost cover all clinical stages from AK to cSCC (*Figure 1A and B*; *Supplementary file 1a*).

We first explored the cellular composition of normal skin. After integration and initial quality control, we acquired single-cell transcriptomes in total of 57,610 cells from all six normal skin samples. Based on the identified variably expressed genes across all normal skin cells, uniform manifold approximation and projection (UMAP) clustering identified nine main clusters. Combining references with Cell-Marker (*Zhang et al., 2019a*), Panglao DB (*Franzén et al., 2019*), Mouse Cell Atlas (*Han et al., 2018*), and ImmGen (*Heng and Painter, 2008*) databases, we annotated each cell population based on their specific markers, including basal cells (COL17A1, KRT5, KRT14), spinous cells (KRT1, KRT10), granular cells (FLG, LOR), proliferating keratinocytes (Pro KCs) (MKI67, TOP2A), follicular cells (KRT6B, KRT17, SFRP1), Langerhans cells (LC) (CD207, CD1A), T cells (CD3D, PTPRC), melanocytes (PMEL, TYRP1), and fibroblasts (DCN, COL1A1) (*Figure 1C and D*).

Notably, we identified different subtypes of basal, spinous, and follicular cells. UMAP analysis classified the keratinocytes into undifferentiated epidermal cells encompassing two subgroups of basal cells (Basal1 and Basal2) and Pro KCs, differentiated keratinocytes encompassing two spinous subpopulations (Spinous1 and Spinous2), and terminally differentiated cells (Granular) (*Figure 1C*). Compared to Basal1, the expression levels of stemness markers of epidermal stem cells in normal skin (COL17A1 [*Matsumura et al., 2016*], TP63 [*Pellegrini et al., 2001*], ITGB1 [*Jones and Watt, 1993*], ITGA3 [*Kurata et al., 2004*] ) were decreased while inflammatory response genes (KRT16, S100A8, S100A9) were increased in Basal2 (*Figure 1D and E*). Functional gene ontology (GO) enrichment analysis of highly expressed genes of Basal1 and Basal2 subpopulations suggested that Basal1 were closely related to hemidesmosomes formation, while Basal2 were related to cell differentiation, migration, and inflammatory response (*Figure 1F*). Thus, we inferred that Basal1 were most likely the quiescent basal cells adhering to the basement membrane, which may represent epidermal stem cells. And Basal2 were cells that have finished proliferation to form the spinous layer for directional differentiation. In two subgroups of spinous cells, Spinous1 were associated with epidermal development, differentiation, and keratinization, while Spinous2 were associated with oxidative phosphorylation, neutrophil degranulation, and immune response (*Figure 1F*). Compared with Spinous1, Spinous2 highly expressed small proline rich region proteins (SPRRs) encoding genes, such as SPRR1B, SPRR2D, and SPRR2E (*Figure 1D and E*), which are involved in the formation of keratinocyte envelope (*Carregaro et al., 2013*). Meanwhile, Spinous2 also highly expressed the cysteine protease inhibitor M/E (CST6), suggesting that Spinous2 subgroup was a well-differentiated upper spinous layer (*Zeeuwen et al., 2001*). Follicular cells were also divided into two groups (Follicular1 and Follicular2). The functional enrichment of Follicular1 suggested that they were related to skin development and differentiation, while Follicular2 showed high levels of genes related to WNT signaling pathway (SFRP1, FRZB,

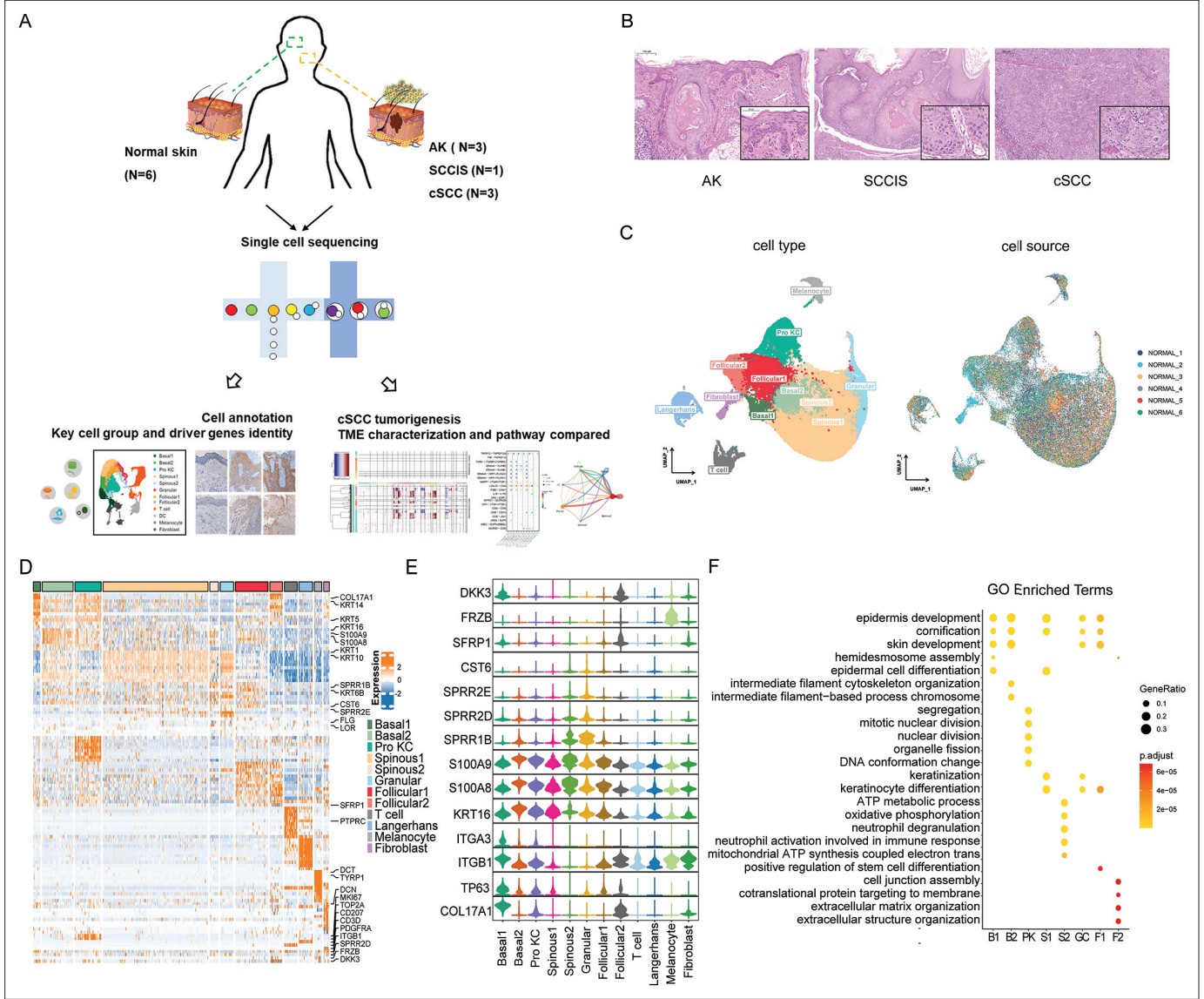

**Figure 1.** Single-cell transcriptome profiling identified different subgroups of keratinocytes in human normal skin. (**A**) Flowchart overview of single-cell sequencing in human skin of actinic keratosis (AK), squamous cell carcinoma in situ (SCCIS), and cutaneous squamous cell carcinoma (cSCC) patients. (**B**) Hematoxylin and eosin (H&E) staining of skin biopsies from representative AK (100× and 250×), SCCIS (50× and 250×), and cSCC (50× and 250×). (**C**) Uniform manifold approximation and projection (UMAP) plot of human normal skin labeled by cell type and patient, respectively. (**D**) Heatmap showing gene expression signatures of each cell type. (**E**) Violin plot displaying the expression of representative genes to identify subpopulations for each cell type. (**F**) Representative gene ontology (GO) terms of signature genes in different cell subpopulations. The color keys from yellow to red indicate the range of p-value.

and DKK3) (*Figure 1D and E*). WNT signaling pathway plays a decisive role in regulating the functions of hair follicle stem cells, and inhibition of WNT signaling pathway can maintain the proliferation and inhibit the differentiation of stem cells (*Lim et al., 2016*). Thus, the Follicular2 may represent outer bulge cells, which have been shown to secrete WNT inhibitors, influencing differentiation of the inner bulge.

In sum, we identified different subgroups in major types of keratinocytes including basal, spinous, and follicular cells. The identification of these subgroups is important to understand the function and mechanisms of keratinocytes in human skin in depth and investigate the origin of cancer cells in the progression from normal skin to cSCC.

## Identification of potential key driver genes from normal skin to AK

To identify the potential drivers for AK, we first performed integration on all AK samples and patient- and site-matched normal samples. UMAP analysis of keratinocytes from AK and its corresponding normal skin samples showed that all AK and normal samples clustering were driven predominantly by cell type rather than by patient. For all three AK samples in this study, the proportion of cell types of each sample was almost the same (*Figure 2A*). These similar cell-type proportion suggested the low individual heterogeneity of AK samples.

Compared to normal group, there was no significant difference in the proportion of basal cells and Pro KCs in AK group (*Figure 2B* and *Figure 2—figure supplement 1*). This inferred that the proliferation and differentiation degree of keratinocytes in AK was not significantly different from that in normal samples. However, the proportion of Follicular1 cells was slightly higher (*Figure 2B* and *Figure 2—figure supplement 1*, t test, p<0.01). It may be related to local epidermal atrophy in AK samples.

To further explore the mechanism of AK at the cell subpopulation level, we identified differentially expressed genes (DEGs) in major cell types of keratinocytes, especially cells with proliferation ability such as basal cells and Pro KCs between AK and normal. 549, 305, and 434 significantly up-regulated DEGs were identified in Basal1, Basal2, and Pro KCs subpopulations, respectively (*Supplementary file 1b-d*, Wilcoxon test, p_val_adj <0.05). GO enrichment analysis showed that it was mainly enriched in the terms associated with epidermal development, oxidative stress response, RNA metabolism, cell cycle, cytoskeleton, response to growth factors, etc. (*Figure 2C*). An analysis between AK-related up-regulated DEGs and genes from the DisGeNET database (*Piñero et al., 2017*) which collected genes and variants associated to human diseases revealed the high correlation of these genes and skin diseases such as dermatologic disorders, dermatitis, atopic, ichthyoses, acanthosis, etc. (*Figure 2—figure supplement 2A–C*).

To identify the key driver genes, we selected top 20 DEGs in major keratinocytes. Combined with functional enrichment analysis and information from published literature, we screened out a group of important candidate genes that may be closely related to AK occurrence and development (*Figure 2D* and *Figure 2—figure supplement 2D*, *Supplementary file 1e*). Among the important candidate genes, some genes have been reported in previous studies showing close relationship with AK or related skin diseases. For example, CDKN2A is well known to take an important role in cSCC, and the latest study also found its mutation in AK (*Thomson et al., 2021*; *Lazo de la Vega et al., 2020*). The up-regulation of CDKN2A in AK and cSCC has already been confirmed in literature (*Inman et al., 2018*). In our study, CDKN2A expression was slightly increased in Basal1 and Basal2 subpopulations in AK stage (avg_log2FC = 0.49 in Basal1, avg_log2FC = 0.36 in Basal2, p_val_adj <0.05), suggesting that CDKN2A may play a key role in the development of AK (*Figure 2—figure supplement 2D*, *Supplementary file 1b and c*). For those genes that have not been reported, we selected seven candidate genes and verified their protein expression levels in an independent cohort including 20 pairs of facial AK and normal skin samples by immunofluorescence (IF). The results showed that the expression of acetaldehyde dehydrogenase 3A1 (ALDH3A1) and insulin-like growth factor binding protein 2 (IGFBP2) was significantly up-regulated in AK tissues and specifically mainly accumulated at the typical keratinocytes of the epidermis (*Figure 2E and F*). Interestingly, there was no significant up-regulation of ALDH3A1 and IGFBP2 in neither SCCIS nor cSCC, and ALDH3A1 even showed down-regulation in cSCC. It indicated the unique role of ALDH3A1 and IGFBP2 in precancerous lesions of skin (*Figure 2—figure supplement 3A and B*).

ALDH3A1, as an important member of the acetaldehyde dehydrogenase superfamily, plays an important role in the occurrence and development of malignant tumors. DEG analysis showed that ALDH3A1 was expressed in almost all keratinocytes and highly expressed especially in Basal1 cells of AK samples (*Figure 2D*). IF experiment showed ALDH3A1 protein expression levels were significantly increased in 85% (17/20) AK tissues (*Figure 2E and F*). Previous study verified that ALDH3A1 has several protective roles, including direct absorption of UV irradiation, scavenging of free radicals, and generation of the antioxidant NADPH in human corneal epithelial cells (*Estey et al., 2007*). In corneal epithelial cells, ALDH3A1-induced reduction in cell growth may contribute to protection against oxidative stress by extending time for DNA and cell repair (*Voulgaridou et al., 2020*). ALDH3A1 is sensitive to UV-induced damage and that UV irradiation leads to inactivation of ALDH3A1. UVB

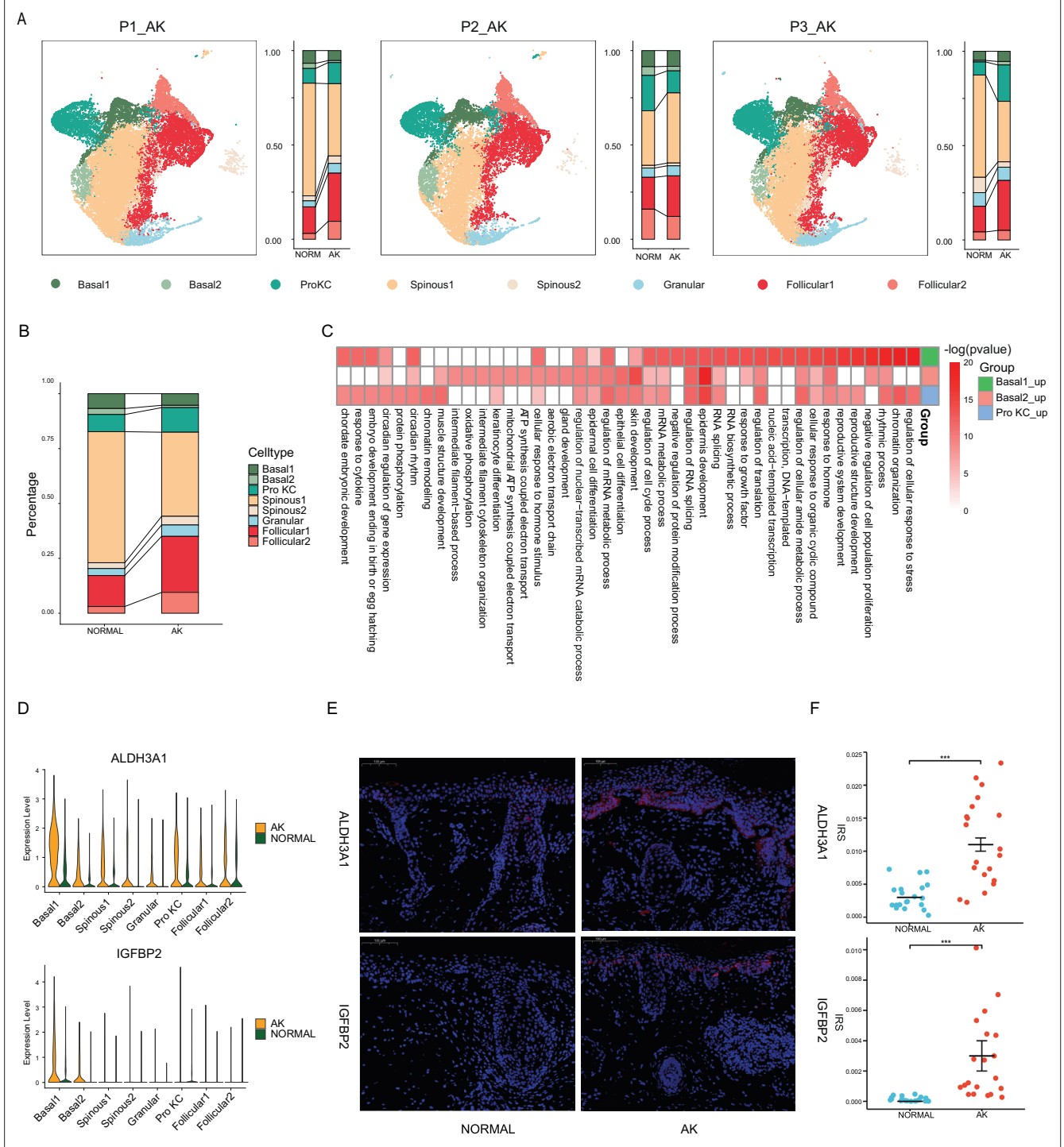

**Figure 2.** Identification of potential key genes driving normal skin to actinic keratosis (AK). (**A**) Uniform manifold approximation and projection (UMAP) of single-cell RNA sequencing (scRNA-seq) cells and cell proportion from each AK patient labeled by cell type. (**B**) Cell proportion of all AK samples and patient-matched normal skin samples. (**C**) Heatmap of gene ontology (GO) terms for up-regulated genes in Basal1, Basal2, and Pro KC subpopulations for AK versus normal skin. (**D**) Violin plots showing the different expression levels of ALDH3A1 and IGFBP2 across cell types in AK and normal samples (Wilcoxon test, p_val_adj <0.05). (**E**) Immunofluorescence staining of ALDH3A1 and IGFBP2 in AK and normal skin groups. Scale bar, 100 μm. (**F**) The mean optical density (IOD/area) analyses of ALDH3A1 and IGFBP2 in AK and normal skin. n=20 for each group. ***p<0.001.

The online version of this article includes the following figure supplement(s) for figure 2:

**Figure supplement 1.** The changes of cell proportion and significance test in all actinic keratosis (AK) samples and patient-matched normal skin samples (t test, p<0.05).

*Figure 2 continued on next page*

*Figure 2 continued*

**Figure supplement 2.** Identification of potential key driver genes from normal skin to actinic keratosis (AK).

**Figure supplement 3.** Expression and functional characterization of ALDH3A1 and IGFBP2.

**Figure supplement 4.** Violin plots showing the expression levels of major factors in Hedgehog signaling pathway across all types of keratinocytes in actinic keratosis (AK) and normal groups (Wilcoxon test).

down-regulates ALDH3A1 expression at the transcriptional and/or post-translational level depending on the dose of UVB in human corneal epithelial cells (*Pappa et al., 2003*).

To explore the relationship between ALDH3A1 expression and UV irradiation, HaCat cells were subjected to UVB irradiation followed by analysis of ALDH3A1 expression. The results showed that ALDH3A1 expression decreased upon UVB irradiation in a dosage-dependent manner (*Figure 2—figure supplement 3C*) in contrast to the increased ALDH3A1 level in AK samples. The discrepancy between in vivo and in vitro results may be attributed to the shorter duration of UVB treatment in vitro. Moreover, the development of AK is a long dynamic process, during which many comprehensive alterations other than short-time UVB irradiation may up-regulate ALDH3A1 expression (*Qu et al., 2020*).

IGFBP2 plays an important role in cell proliferation, differentiation, apoptosis, and epithelial mesenchymal transition (EMT). The high expression of IGFBP2 is significantly correlated with the malignant progression and prognosis of melanoma (*Li et al., 2020*) and other tumors. In epidermal cells, recent study reported the aberrant expression of IGFBP2 and its dual action during growth and senescence processes in psoriatic keratinocytes. Intracellular IGFBP2 can inhibit apoptosis by interacting with p21 and abrogation of IGFBP2 leads to the restoration of common apoptosis mechanisms in psoriatic keratinocytes (*Mercurio et al., 2020*). In our study, IGFBP2 was specially highly expressed in Basal1 and Basal2 cells of AK samples (*Figure 2D*). IF experiment showed that IGFBP2 protein expression levels were significantly increased in 75% (15/20) AK tissues (*Figure 2E and F*). It is inferred that in AK development, IGFBP2 might also take an important role in keratinocytes. To investigate the effects of IGFBP2 on epidermal cells, we performed overexpression of IGFBP2 in HaCaT and A431 cells and measured the cell proliferation and cell invasion. The results showed that the overexpression of IGFBP2 promoted the proliferation of HaCaT cells and A431 cells, while also promoting the invasive ability (p<0.05, *Figure 2—figure supplement 3D and E*).

Collectively, these results indicated that from normal skin to AK, a lot of genes have changed expression. Especially, the basal cells specific up-regulated molecules ALDH3A1 and IGFBP2 likely contribute to the development of AK and may be the key driver genes in the process from photoaged skin to AK.

## Monotonically changed DEGs play important roles in the progression of AK to SCCIS

The individual P2 with both AK and SCCIS is a typical model to investigate the mechanisms of development from AK to SCCIS. We first integrated AK sample, SCCIS sample, and normal adjacent skin sample from P2. In order to capture enough keratinocytes in AK samples for subsequent research, we separated the epidermal tissue from the dermis. In SCCIS sample, the dermis was not separated, therefore it contained more non-keratinized cells including endothelial cells, T cells, vascular smooth muscle cells (VSMC), fibroblasts, etc. (*Figure 3A*). The marker genes COL17A1, MKI67, and KRT1 represent basal, Pro KC, and terminally differentiated cells, respectively. The expression of these known representative genes showed the distribution of corresponding cell types (*Figure 3A*). Notably, the proportion of basal cells in SCCIS is significantly increased compared with normal and AK samples, suggesting their specificity in SCCIS (*Figure 3B*).

To identify DEGs that monotonically increased during the process from normal skin to AK and SCCIS, which could be associated with the transformation from precancerous lesions to cancer, we first obtained AK up-regulated genes compared to normal, and SCCIS up-regulated genes compared to AK respectively in basal subpopulation, then got the overlap of these two gene sets. There are 21 overlapped up-regulated genes (*Figure 3C* and *Supplementary file 1f*), most of them were reported to have important functions in skin disease including cSCC. For example, the growth-controlling transcription factor Kruppel-like factor 6 is an important contributor for epidermal decline and aging (*Figure 3D*; *Zou et al., 2021*). The activator protein-1 (AP-1) family transcription factor subunit gene

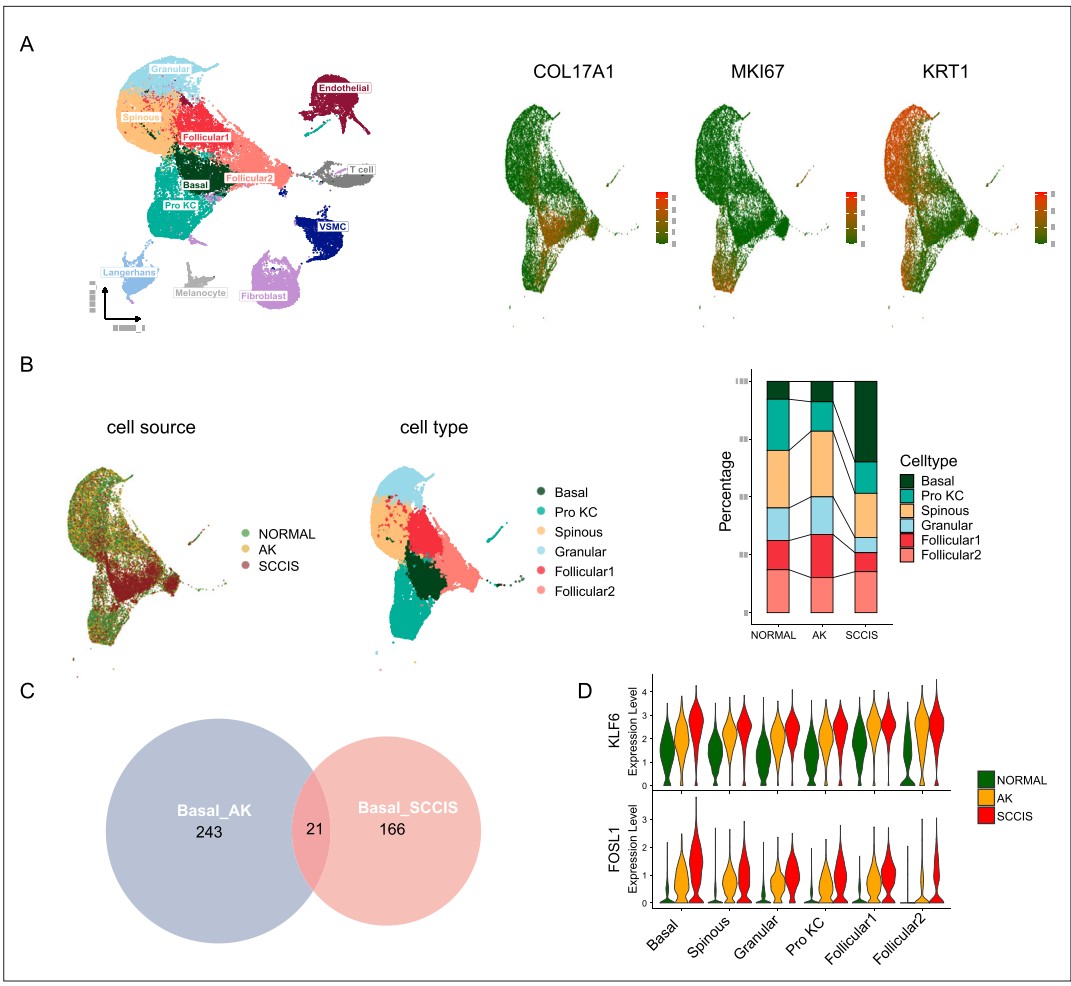

**Figure 3.** Monotonically changed differentially expressed genes (DEGs) play important roles in the progression of actinic keratosis (AK) to squamous cell carcinoma in situ (SCCIS). (**A**) Left, uniform manifold approximation and projection (UMAP) of all cells from patient (**P2**) with both AK and SCCIS labeled by cell types; right, expression of basal, Pro KC, and differentiated genes in all keratinocytes from P2. (**B**) Left, UMAP of all keratinocytes from P2 labeled by patient and cell type, respectively; right, cell proportion of normal, AK, and SCCIS samples in P2. (**C**) Overlap of up-regulated genes in basal cells from AK compared to normal and SCCIS compared to AK. (**D**) Violin plots showing the different expression levels of KLF6 and FOSL1 in normal, AK, and SCCIS samples in P2 (Wilcoxon test, p_val_adj <0.05).

FOS-like 1 (FOSL1) is considered to be the potential driver of transformation from SCCIS to cSCC and is selectively highly expressed at the frontier of invasion in cSCC tumor cells (*Figure 3D*; *García-Díez et al., 2019*). Another AP-1 family transcription factor subunit JunD proto-oncogene (JUND) can regulate cell proliferation, differentiation, and apoptosis. It has been proposed to protect cells from p53-dependent senescence and apoptosis (*Weitzman et al., 2000*). These data indicated that the constant up-regulation of these key growth-controlling transcription factors may play important roles in the process of AK to SCCIS. In addition, the ubiquitin binding protein sequestosome 1 (SQSTM1) is involved in cell signal transduction, oxidative stress, and autophagy. It was verified that UVA- induced SQSTM1 could act through COX-2 to promote the growth and progression of skin tumors (*Sample et al., 2017*). The Ras homolog family member B (RHOB) is a key regulator of UVB response. UVB-induced RHOB overexpression is involved in the initiation of cSCC by promoting the survival of keratinocytes with DNA damage mutations (*Meyer et al., 2014*). All of these findings suggested that these monotonically up-regulated genes are potential key drivers from precancerous lesion to carcinoma in situ during the development of cSCC, which may become potential targets for the prevention and treatment of cSCC.

In addition, we also investigated the monotonically down-regulated DEGs in this individual (*Supplementary file 1g*). Although the number of these genes is also small, many of them showed strong associations with skin disorder or cancer, especially with cancer suppression. For instance, DNA damage-inducible transcript 4 regulates apoptosis in response to DNA damage via its effect on mammalian target of rapamycin complex 1 activity (*Ip et al., 2017*). It is associated with skin atrophy (*Agarwal et al., 2019*). The S100A family member S100A14 can regulate cell survival and apoptosis by modulating TP53 expression (*Chen et al., 2012*). Levels of S100A14 have been found to be lower in cancerous tissue and associated with metastasis, suggesting a tumor suppressor function (*Chen et al., 2009*). Enolase 1 has been shown to bind to the c-myc promoter and function as a tumor suppressor (*Feo et al., 2000*). Besides, the SPRR family member SPRR2A also showed constant down-regulation. Recent study has identified it as a noninvasive biomarker in gastric cancer (*Xu et al., 2020*). Considering it is the component of cornified keratinocyte cell envelope and a well-known keratinocyte terminal differentiation marker (*Fischer et al., 1998*), we have reason to speculate the constant down-regulation of SPRR2A, indicating cancerization and increased invasiveness during the development from AK to SCCIS.

Although the underlying mechanisms of these genes in carcinogenesis need further functional studies, all the above data provided abundant evidences that these monotonically up-regulated and down-regulated DEGs act synergistically and play key driving roles in the formation of SCCIS from AK.

## Identification of malignant basal subpopulation in SCCIS

The increased proportion of basal cells in SCCIS hinted that they might be the crucial cell types in carcinomatous change of AK, thus we first investigated the characteristics of these cells. GO enrichment analysis showed that signature genes of basal cells in SCCIS were closely related to the biological processes of cell proliferation, morphological change, migration, cell connection, and extracellular matrix (ECM), suggesting their invasive behavior (*Figure 4A*). To define malignant cells, we calculated large-scale chromosomal copy number variation (CNV) in each cell type of keratinocytes based on averaged expression patterns across intervals of the genome. We found that a subgroup of basal cells in SCCIS exhibited remarkably higher CNV levels than other basal cells (*Figure 4B*). UMAP analysis of retrieved basal cells showed that basal cells in SCCIS were divided into two major subgroups and basal cells with higher CNV levels were almost enriched in one subgroup (*Figure 4C*).

The presence of these two different subpopulations of basal cells in SCCIS prompted us that they had different malignant status. We named the basal cells in SCCIS with higher CNV levels as Basal-SCCIS-tumor cells and the subgroup with lower CNV levels as Basal-SCCIS-normal cells. We next focused on the gene expression patterns in these two subpopulations and identified a total of 238 up-regulated genes in Basal-SCCIS-tumor cells (*Figure 4D* and *Supplementary file 1h*). GO and Kyoto Encyclopedia of Genes and Genomes (KEGG) enrichment analysis revealed that these genes were mainly associated with neutrophil degranulation, protein folding, keratosis, hemidesmosome assembly, cell proliferation, apoptotic signaling pathway, hemopoiesis, myeloid cell differentiation, stress response, and cell junction organization (*Figure 4E*). Notably, in Basal-SCCIS-tumor cells, DNA damage response-related replication genes (PCNA, MCM7) were significantly up-regulated (*Figure 4F*). Especially MCM7 is required for S-phase checkpoint activation upon UV-induced damage, which indicated that the Basal-SCCIS-tumor cells were malignant cells from AK induced by UV damage. In addition, a large number of heat shock protein (HSP)-related genes (HSPA1A/B, HSP90AA1, HSPA6) were highly expressed in Basal-SCCIS-tumor cells, as well as activated keratin genes (KRT6A/B/C, KRT16, KRT17, KRT19) and S100 family genes (S100A7, S100A8, S100A9) (*Figure 4G and H*). HSPs play a role in tumor-related biological processes such as cell proliferation, apoptosis, invasion, tumor cell stemness, angiogenesis, glycolysis, hypoxia, and inflammation. The HSP family is considered as a promising target for anticancer therapy. UV irradiation can induce keratinocyte injury and significant up-regulation of HSPs of in vitro skin model (*Marionnet et al., 2014*), which was consistent with our results (*Figure 4H*). Previous studies have recognized that activated keratins are key early barrier alarmins, and the up-regulation of these genes are involved in the alteration of proliferation, cell adhesion, migration, and inflammatory characteristics of keratocytes, leading to hyperproliferation and innate immune activation of keratocytes in response to epidermal barrier disruption (*Zhang et al., 2019b*). The S100A family members were also reported to be significantly up-regulated in skin disorders or epithelial skin tumors. They participate in the immunoreactivity of keratocytes and have a potential

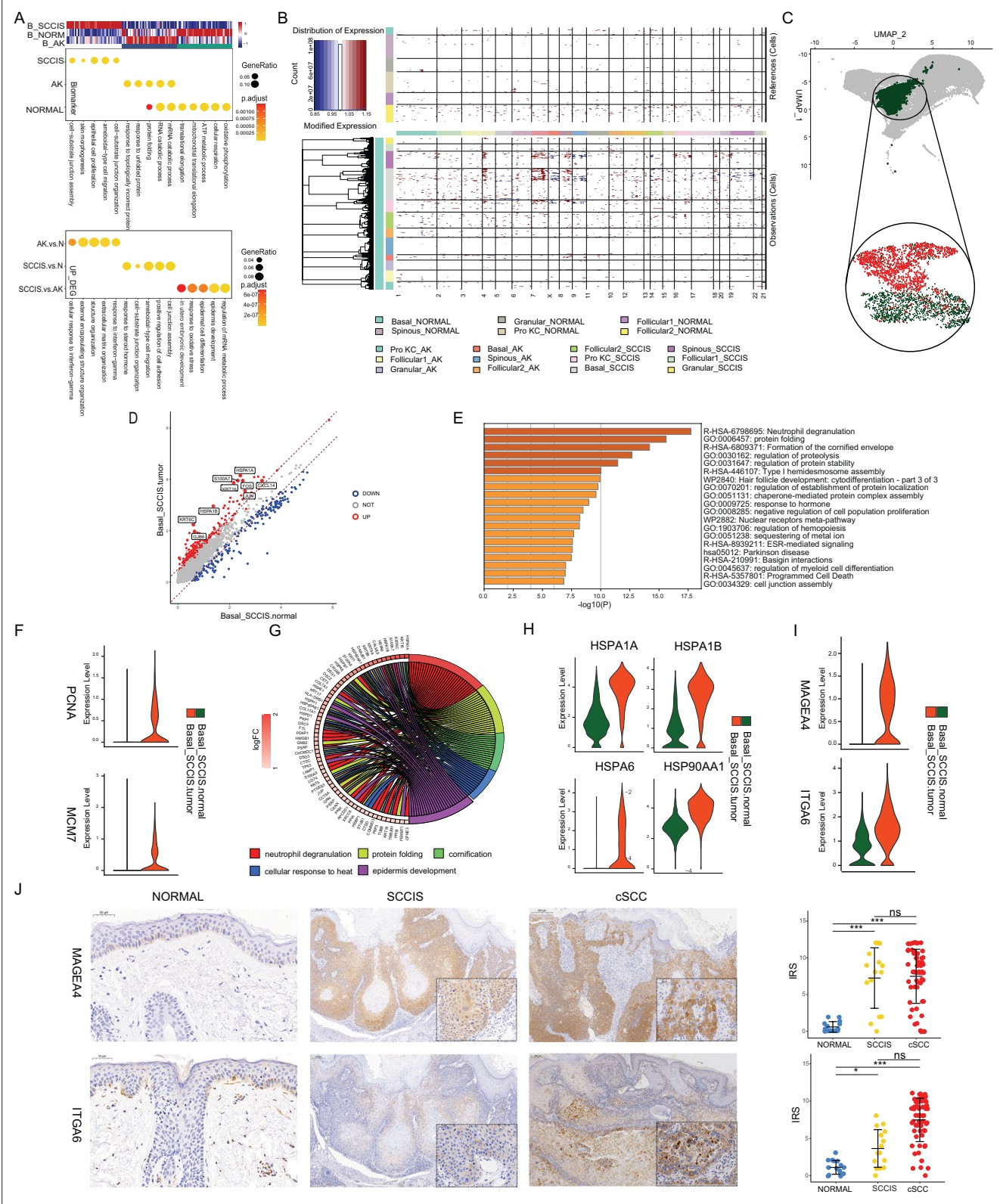

**Figure 4.** Identification of malignant basal subpopulation in squamous cell carcinoma in situ (SCCIS). (**A**) Representative gene ontology (GO) terms for genes with specific expression in basal cell of SCCIS, actinic keratosis (AK), and normal samples in P2 (upper) and up-regulated differentially expressed genes (DEGs) from AK versus normal, SCCIS versus normal, SCCIS versus AK, respectively (lower). (**B**) Heatmap showing copy number variation (CNV) levels of all keratinocytes from AK and SCCIS samples in P2. The keratinocytes from normal sample in P2 were defined as references. (**C**) Uniform

*Figure 4 continued*

manifold approximation and projection (UMAP) of subgroups generated from basal cells in SCCIS sample showing basal cells with higher CNV level enriched in one subgroup; red dot representing basal cells with higher CNV level (cnv.score>81.5). (**D**) DEGs detected between Basal-SCCIS-tumor and Basal-SCCIS-normal. (**E**) Representative enriched Kyoto Encyclopedia of Genes and Genomes (KEGG) and GO terms in up-regulated genes (avg_log$_2$FC >0.58 and p_val_adj <0.05). (**F**) Violin plots showing the expression level of representative DNA damage response marker genes in Basal-SCCIS-tumor and Basal-SCCIS-normal subgroups. (**G**) Chord plot showing the top up-regulated genes included in representative GO terms. (**H**) Violin plots showing the expression level of major members in HSP family across Basal-SCCIS-tumor and Basal-SCCIS-normal subgroups (Wilcoxon test, p_val_adj <0.05). (**I**) Violin plots showing the expression level of MAGEA4 and ITGA6 in Basal-SCCIS-tumor and Basal-SCCIS-normal subgroups (Wilcoxon test, p_val_adj <0.05). (**J**) Immunohistochemistry staining of MAGEA4 and ITGA6 in human skin of normal, SCCIS, and cutaneous squamous cell carcinoma (cSCC) samples.

The online version of this article includes the following figure supplement(s) for figure 4:

**Figure supplement 1.** The comprehensive analysis of the patient (**P2**) with both actinic keratosis (AK) and squamous cell carcinoma in situ (SCCIS).

**Figure supplement 2.** The co-expression of MAGEA4, COL17A1, and PCNA in squamous cell carcinoma in situ (SCCIS).

**Figure supplement 3.** Expression and functional characterization of MAGEA4 and ITGA6.

**Figure supplement 4.** Comparison of basal cells with the cell populations defined by Ji et al.

utility as biomarkers for cancerous malignancies (*Halawi et al., 2014*). Taken together, Basal-SCCIS-tumor cells with high CNV level may be highly invasive. As malignant cells, they may migrate and invade the dermis, and develop into invasive cSCC by promoting the proliferation ability of cells and the degradation of ECM to destroy the basement membrane.

## Basal-SCCIS-tumor-specific genes were closely associated with progression from SCCIS to cSCC

Besides those genes were already reported closely related to cancerous malignancies of SCCIS, we identified a group of candidate genes in the up-regulated genes in Basal-SCCIS-tumor subgroup that are closely related to tumor development. Combing with the single-cell transcriptomic data from invasive cSCC samples, we further screened out the candidate genes that were not only highly expressed in SCCIS samples, but also highly expressed in keratinocytes with proliferative capacity or differentiative potential (Basal1, Basal2, Pro KC, and Follicular1) in cSCC tumor samples, which play an important role in the progression of SCCIS to invasive cSCC. Comprehensively considering the fold change of expression and information from published literatures, we selected the candidate genes for further validation (*Figure 4I* and *Figure 4—figure supplement 1A*, *Supplementary file 1i*). These candidate genes were validated by immunohistochemistry (IHC) in an independent set of samples, including 15 normal skin tissues, 15 SCCIS tissues, and 60 invasive cSCC tissues (36 well-differentiated cSCC samples, 24 moderately differentiated/poorly differentiated cSCC samples), all of which were obtained from the faces of elderly patients.

Among them, MAGE family member A4 (MAGEA4) was found to be strongly positive in most SCCIS (73.33%) and invasive cSCC (76.67%). The expression of MAGEA4 in tumor group was significantly higher than that in normal group (p<0.001, *Figure 4J*). However, the expression of MAGEA4 was not significantly different among SCCIS, well-differentiated cSCC group and moderately differentiated/poorly differentiated cSCC group (*Figure 4—figure supplement 1B*), suggesting MAGEA4 might be activated continuously in SCCIS and cSCC tumors of all stages. MAGEA4 was found to be highly expressed in melanoma, pancreatic cancer, lung cancer, and esophageal squamous cell carcinoma (*Tang et al., 2016*). Considering its potential utility as an indicator for malignancies of SCCIS tumor cells, we performed IF co-localization of COL17A1, PCNA, and MAGEA4 in SCCIS tissues to investigate stemness and proliferative state of tumor cells. We found that in MAGEA4+ tumor cells of SCCIS, the stem cell marker COL17A1 and the proliferation marker PCNA were significantly increased compared with the adjacent tissues (*Figure 4—figure supplement 2*). Besides, recent study has also identified it as a biomarker of tumor-specific keratinocyte (TSK), a special subgroup of basal tumor cells in cSCC, verifying its close relationship with invasive cSCC (*Ji et al., 2020*). Taken together, we inferred that MAGEA4 might become a new biomarker of higher malignancy in certain SCCIS individuals.

In addition, tumor-related gene integrin subunit alpha 6 (ITGA6) was also significantly overexpressed in Basal-SCCIS-tumor cells. The IHC results showed that ITGA6 only scattered expression in the basal

layer of the normal skin tissue and was up-regulated in SCCIS (p<0.05) and invasive cSCC (p<0.001), as well as being expressed at a higher level in invasive cSCC than in SCCIS (p<0.001, *Figure 4J*). However, there was no significant difference of ITGA6 expression between the well-differentiated cSCC group and moderately differentiated/poorly differentiated cSCC group (*Figure 4—figure supplement 1C*). We observed moderate to strong cytoplasmic and membranous positivity of ITGA6 in tumor cells. ITGA6 expression can indicate the progenitor potential of mesenchymal stem cells (*Nieto-Nicolau et al., 2020*). Recent studies have reported that high ITGA6 expression enhances invasion and tumor-initiating cell activities in metastatic breast cancer (*Brooks et al., 2016*), providing evidence for the value of ITGA6 as cancer stem cell marker.

To further confirm the important role of MAGEA4 and ITGA6 in the development from SCCIS to cSCC, we performed functional experiment in human immortalized keratinocytes (HaCaT) and cSCC cell lines (A431, SCL-I, SCL-II). We first investigated the expression levels of MAGEA4 and ITGA6 in these cell lines and observed that the expression of MAGEA4 mRNA was extremely high in A431 cells, but not detectable in SCL-I and SCL-II cells (*Figure 4—figure supplement 3A*), which is consistent with the immunohistochemical results observed in our clinical samples and *Muehleisen et al., 2007*. We silenced the expression of MAGEA4 gene in A431 cells by small interfering RNA (siRNA) (*Figure 4—figure supplement 3B*), and the results showed that the proliferation, migration, invasive ability of A431 cells was significantly reduced (p<0.01), while the apoptosis rate was increased (p<0.01, *Figure 4—figure supplement 3C*). The silencing of ITGA6 also significantly reduced the ability of proliferation, migration, and invasion in the three cSCC cell lines (p<0.01, *Figure 4—figure supplement 3B, D, F, and G*), while apoptosis was significantly increased (p<0.01, *Figure 4—figure supplement 3E*). It is suggested that MAGEA4 and ITGA6 had a potential carcinogenic role in the progression of SCCIS to cSCC by regulating cell stemness, proliferation, apoptosis, and ECM degradation (*Figure 4—figure supplement 3*).

## CNV scores positively correlated with malignant degrees of cSCC

To deeply investigate the genesis and key drivers of cSCC, we first integrated all three cSCC tumors and patient- and site-matched normal skin (*Figure 5A*). These three individuals represent different malignant degrees of cSCC (*Supplementary file 1a*). The cell-type proportion analysis and the expression of cell proliferation and differentiation marker genes reflected significant difference between tumor and normal samples (*Figure 5B and C*). Basal1 cells showed higher proportion in tumor samples of each patient compared to normal skins, which indicated the loss of terminal differentiation in tumor basal cells (*Figure 5A–C* and *Figure 5—figure supplement 1*, t test, p<0.05). To define malignant cells, we identified large-scale CNV of keratinocytes based on averaged expression patterns across intervals of the genome. Keratinocytes in normal samples were set as reference cells. The presence of CNV in cSCC samples suggests that these cells may be tumor cells in squamous cell carcinoma samples. These tumor cells were mainly derived from Basal, Pro KCs, Follicular2 cells, and a small number of spinous cells (*Figure 5—figure supplement 2A*). In poorly differentiated cSCC sample, the significant gains in chromosomes 7, 9 and deletion in chromosome 10 were reported in the previous study, indicating the reliability of the CNV analysis results (*Figure 5—figure supplement 2C*; *Purdie et al., 2009*). Then, we compared the CNV landscapes among three patients. Well-differentiated individual displayed low CNV scores (*Figure 5—figure supplement 2B*). In contrast, poorly differentiated cSCC individual exhibited remarkably higher CNV levels in most types of keratinocytes (*Figure 5—figure supplement 2C*), while moderately differentiated individual had moderate CNV levels (*Figure 5—figure supplement 2D*). This indicated that there was significant heterogeneity among different cSCC individuals and the CNV levels of individuals could reflect their malignant status.

## Identification and functional characterization of key genes associated with cSCC

In order to understand the gene expression profile characteristics of invasive cSCC, we identified and analyzed gene function enrichment of significantly up-regulated DEGs in important cell subpopulations of cSCC compared to normal skin tissues. 778, 1044, 1159, and 760 significantly up-regulated DEGs were identified among Basal1, Basal2, Pro KCs, and Follicular2 cell subpopulations, respectively (*Supplementary file 1j-m*, Wilcoxon test, p_val_adj <0.05). GO analysis was mainly concentrated in various tumor-related biological processes such as cell morphological change and adhesion,

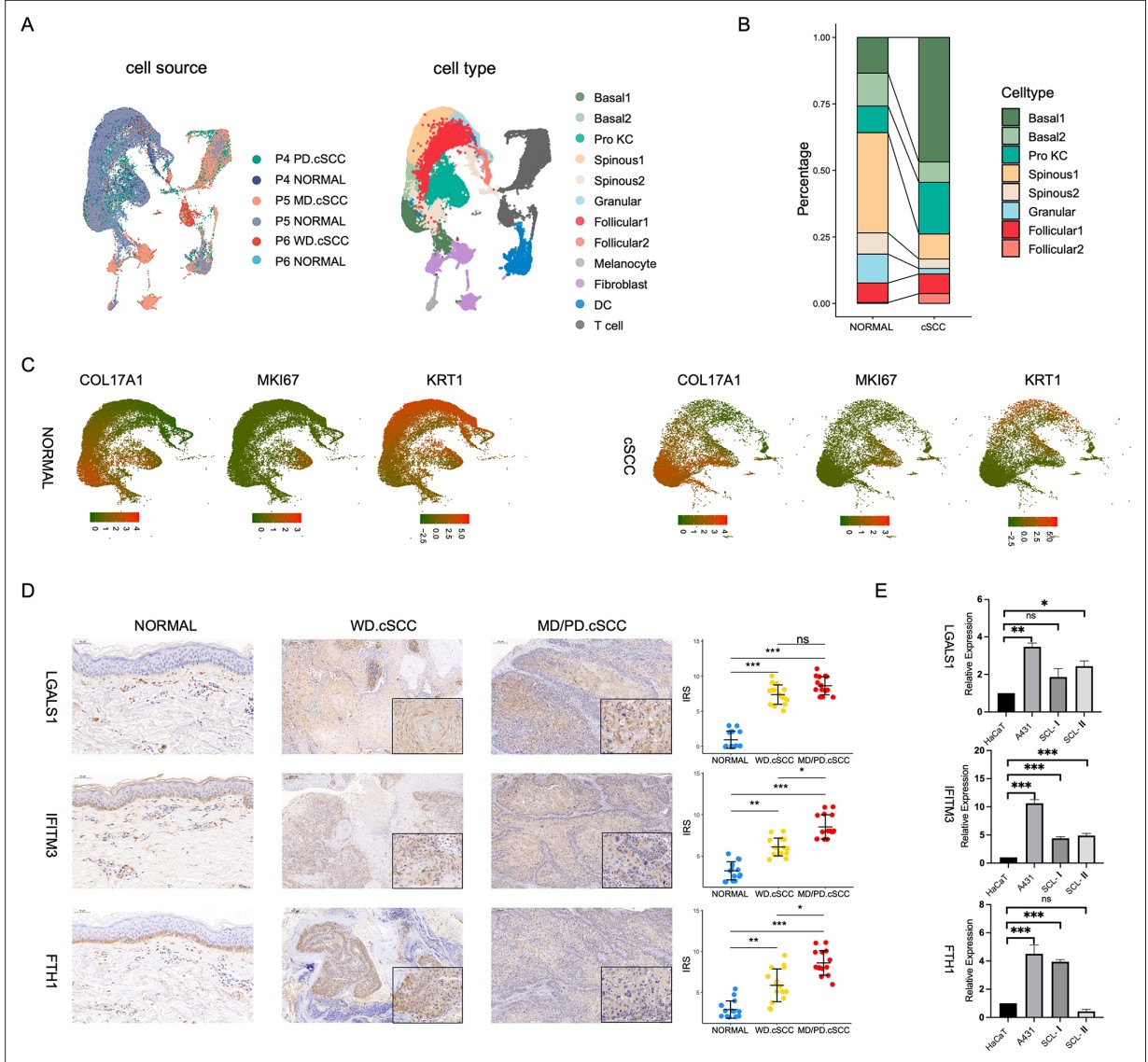

**Figure 5.** Identification of key genes associated with cutaneous squamous cell carcinoma (cSCC). (**A**) Uniform manifold approximation and projection (UMAP) of all cells from cSCC patients labeled by sample and cell type, respectively. (**B**) Cell proportion of keratinocytes in cSCC and normal groups. (**C**) Expression of basal, Pro KC, and differentiated genes in all keratinocytes of cSCC and normal groups. (**D**) Left, immunohistochemical staining of LGALS1, IFITM3, and FTH1 in normal skin (200×), well-differentiated cSCC (WD cSCC) (50× and 250×), and moderately differentiated/poorly differentiated cSCC (MD/PD cSCC) (50× and 250×). Scale bar, 200 μm and 50 μm. Right, the immunoreactivity score (IRS) analyses of LGALS1, IFITM3, and FTH1 in normal skin, WD cSCC, and MD/PD cSCC. n=15 for each group. *p<0.05; **p<0.01; ***p<0.001; ns, not significant. (**E**) The mRNA expression of LGALS1, IFITM3, and FTH1 in human immortalized keratinocytes (HaCaT) and cSCC cell lines (A431, SCL-I, SCL-II). *p<0.05; **p<0.01; ***p<0.001; ns, not significant.

The online version of this article includes the following figure supplement(s) for figure 5:

**Figure supplement 1.** The changes of cell proportion and significance test in all cutaneous squamous cell carcinoma (cSCC) samples and patient-matched normal skin samples (t test, p<0.05).

**Figure supplement 2.** Copy number variation (CNV) scores positively correlated with malignant degrees of cutaneous squamous cell carcinoma (cSCC).

**Figure supplement 3.** Identification of key genes associated with cutaneous squamous cell carcinoma (cSCC).

signaling pathway regulation, apoptosis and angiogenesis, as well as processes related to immunity including antigen processing and presentation, myeloid cell differentiation, and negative regulation of immune response, etc. (*Figure 5—figure supplement 3A*). There were 888 and 247 significantly up-regulated DEGs in Spinous1 and Spinous2 cell subpopulations, respectively (*Supplementary file 1n, o*, Wilcoxon test, p_val_adj <0.05). In addition to the biological process similar to basal cells,

GO enrichment analysis showed that DEGs were mainly enriched in the regulation of programmed death, ATP metabolism, and the production of type I interferon (*Figure 5—figure supplement 3A*). Based on above differential gene expression and functional enrichment analysis, we identified a group of important candidate genes that may be closely related to tumor genesis and development in cSCC including CD74, CDKN2A, COL17A1, JUND, MMP1, BST2, LGALS1, IFITM3, ISG15, IFI6, FTH1, LAMA3, LAMC2, SAT1, and so on (*Figure 5—figure supplement 3B*).

To verify the expression of those potential key genes with important functions in cSCC, we first performed IHC experiment of these genes with significant differences in independent cohort including 30 cases of facial cSCC (15 well-differentiated cSCC samples and 15 moderately differentiated/poorly differentiated cSCC samples) and 15 cases of para-cancer normal skin tissues. Among them, three out of eight genes were verified that they had significantly higher expression in cSCC group compared to normal group (*Supplementary file 1p*). It was found that the protein expression levels of galectin 1 (LGALS1), interferon-induced transmembrane protein 3 (IFITM3), and ferritin heavy chain 1 (FTH1) genes were significantly increased in cSCC (*Figure 5D*). As a key promoter of angiogenesis and fibrosis, LGALS1 inhibits tumor immune response and is highly expressed in melanoma and head and neck cancer (*Chawla et al., 2016*). In this study, LGALS1 showed moderate to strong cytoplasmic and nuclear immunoreactivity in most well-differentiated and poorly differentiated cSCC tumor cells, and some of them were weak staining, while para-cancer normal skin epidermal keratinocytes were almost negative. The LGALS1 expression in the tumor groups was significantly higher than that in the normal group (p<0.001), but there was no statistical significance between the well-differentiated and poorly differentiated cSCC groups (*Figure 5D*). IFITM3 is an interferon-stimulating response-related gene, which is related to cell proliferation, cell cycle regulation, autophagy, inflammation, EMT, and many other processes. FTH1 is associated with iron metabolism, and may be involved in the protection of DNA from oxidative damage as well as in the regulation of inflammation and tumor immune microenvironment. Although IFITM3 and FTH1 showed moderate cytoplasmic/membranous immunoreactivity in basal cells in normal tissues, they were negatively expressed in other keratinocytes. In poorly differentiated cSCC, IFITM3 and FTH1 showed moderate to strong cytoplasmic, membranous immunoreactivity and a small number of nuclear staining. Medium to strong staining can also be seen at the leading edge or poorly differentiated keratinocytes of the well-differentiated cSCC, while the expression is negative/weakly positive in the differentiated keratinocytes and the central keratinized areas of tumors (*Figure 5D*). Overall, the expression of IFITM3 and FTH1 in poorly differentiated cSCC was significantly higher than that of normal group (p<0.001) and well-differentiated group (p<0.05) (*Figure 5D*). We also verified the mRNA expression levels of LGALS1, IFITM3, FTH1 in human immortalized keratinocytes (HaCaT) and human cSCC cell lines (A431, SCL-I, SCL-II). The results showed that these genes were significantly overexpressed in at least two human cSCC cell lines compared to HaCaT (*Figure 5E*). Besides, although the bone marrow stromal cell antigen 2 (BST2) and spermine N1-acetyltransferase 1 (SAT1) genes showed weak immunoreactivity in normal skin tissues and there were no statistically significant differences between cSCC and normal group (*Figure 5—figure supplement 3C*), the mRNA expression levels of BST2 and SAT1 were significantly overexpressed in human cSCC cell lines compared to HaCaT (*Figure 5—figure supplement 3D*).

To investigate the effects of LGALS1, IFITM3, FTH1, BST2, and SAT1 on the proliferation of human cSCC cells, the human cSCC cell lines A431, SCL-I, and SCL-II were transfected with siRNA targeting these genes (*Figure 6A* and *Figure 6—figure supplement 1A*). The results showed that the silencing of LGALS1, IFITM3, FTH1, BST2, and SAT1 genes all inhibited the proliferation of the three human cSCC cells to varying degrees (*Figure 6B* and *Figure 6—figure supplement 1B*). These results suggested that LGALS1, IFITM3, FTH1, BST2, and SAT1 could regulate cell proliferation in cSCC. Then, Annexin V-FITC/propidium iodide (PI) staining and flow cytometry (FCM) was applied to quantify the effect of genes on apoptosis in human cSCC cells. The results showed that the gene silencing significantly increased the apoptosis rate of A431, SCL-I, and SCL-II tumor cells (p<0.01) (*Figure 6C* and *Figure 6—figure supplement 1C*). These results suggest that the up-regulation of LGALS1, IFITM3, FTH1, BST2, and SAT1 in cSCC may inhibit the apoptosis of tumor cells.

The effect of gene interference on the migration ability of human cSCC cells was detected by cell scratch assay. The results showed that after knocking down LGALS1, IFITM3, BST2, and SAT1, the migration distances of A431, SCL-I, and SCL-II tumor cells were reduced after 72 hr of scratching compared with the control group (p<0.01); the tumor cells' migration ability was decreased, while

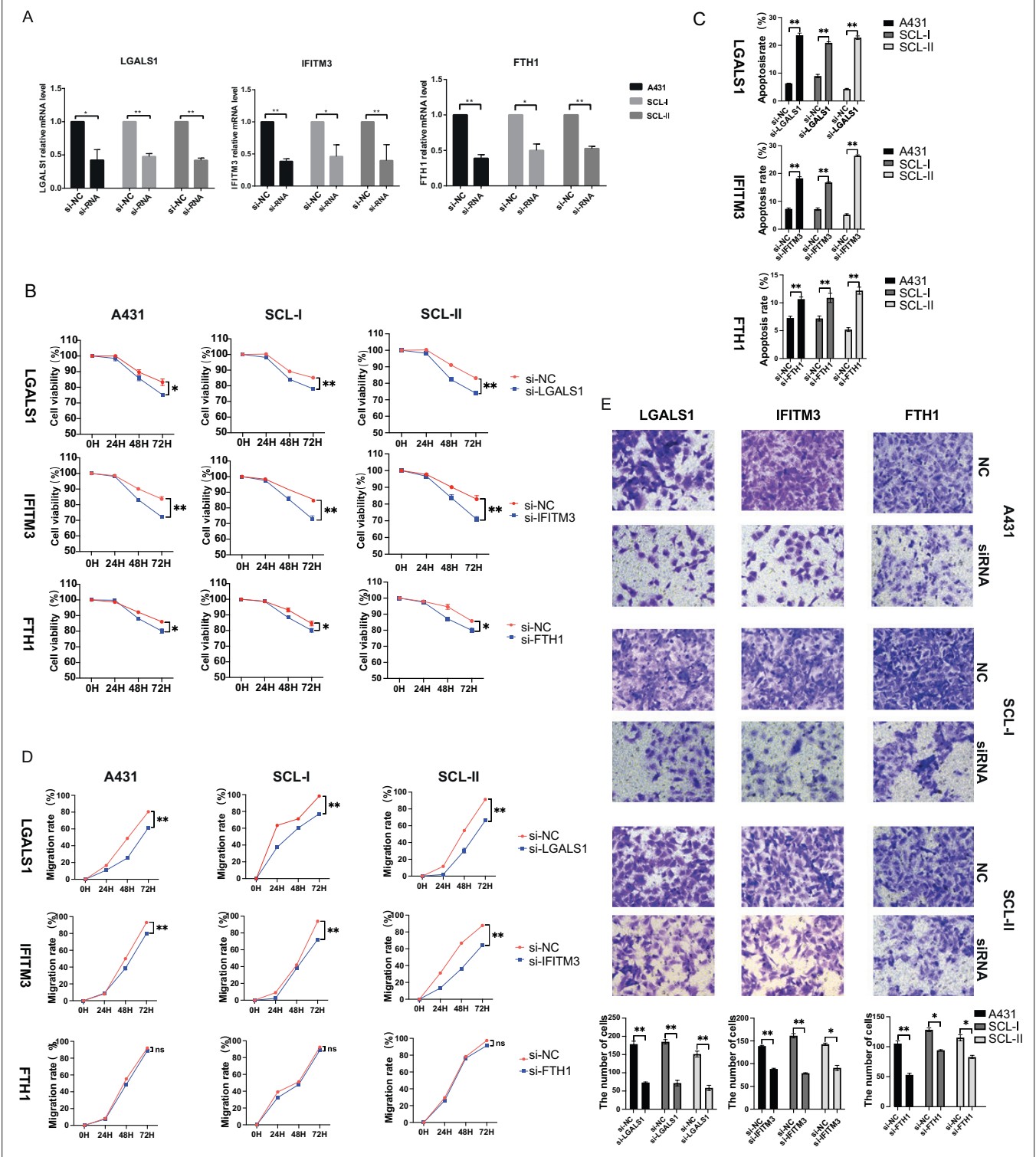

**Figure 6.** Functional characterization of key genes associated with cutaneous squamous cell carcinoma (cSCC). (**A**) Effect of small interfering RNA (siRNA) on the expression of LGALS1, IFITM3, and FTH1 in A431, SCL-I, and SCL-II determined by quantitative real-time PCR (qRT-PCR). (**B**) Effect of LGALS1, IFITM3, and FTH1 on cSCC cell proliferation. The CCK-8 proliferation assay demonstrated a significant decrease in the proliferation of the si-LGALS1, si-IFITM3, and si-FTH1 groups compared with the si-NC group. *p<0.05; **p<0.01; ***p<0.001; ns, not significant. (**C**) The effect of LGALS1, IFITM3, and FTH1 on cSCC cell apoptosis. Significant increase in the apoptosis of the si-LGALS1, si-IFITM3, and si-FTH1 groups compared with the si-NC group. *p<0.05; **p<0.01; ***p<0.001; ns, not significant. (**D**) The scratch experiment showed that LGALS1 and IFITM3 knockdown resulted in a shorter vertical migration distance compared with the control group after 72 hr, while there was no significant change in the si-FTH1 group. *p<0.05;

*Figure 6 continued on next page*

*Figure 6 continued*

**p<0.01; ***p<0.001; ns, not significant. (**E**) Transwell assay showed that the invasion abilities of the si-LGALS1, si-IFITM3, and si-FTH1 groups significantly decreased compared with the si-NC group. *p<0.05; **p<0.01; ***p<0.001.

The online version of this article includes the following figure supplement(s) for figure 6:

**Figure supplement 1.** Expression and functional characterization of BST2 and SAT1.

there was no significant difference with FTH1 in any tumor cells (p>0.05) (*Figure 6D* and *Figure 6—figure supplement 1D*). These results suggest that up-regulation of LGALS1, IFITM3, BST2, and SAT1 may promote tumor cell migration ability in cSCC, whereas FTH1 has no great effect on this. Transwell invasion assay was used to detect the effect of gene interference on the invasion ability. The results showed that LGALS1, IFITM3, FTH1, BST2, and SAT1 gene silencing significantly reduced the invasion ability of A431, SCL-I, and SCL-II (*Figure 6E* and *Figure 6—figure supplement 1E*). All these results suggested that LGALS1, IFITM3, FTH1, BST2, and SAT1 may take an important role in cSCC by regulating the processes of cell proliferation, apoptosis, migration, and invasion.

## The TME landscape of cSCC

The progression of AK to cSCC is not only cancerization of keratinocytes, but also closely related to changes in skin microenvironment (*Nissinen et al., 2016*). Long-term UVB irradiation can cause epidermal cell damage, while UVA can reach dermis and cause activation and oxidative damage of various cells in dermis (*Taguchi et al., 2013*). For example, Langerhans may have disorders in cell number, migration ability, phenotypic changes, and antigen presentation ability (*Taguchi et al., 2013*). In addition, studies have confirmed fibroblast activation and expression of macrophage (Mac) proteinases in the matrix, as well as loss of collagen XV and XVIII from the dermal basement membrane are early events in the progression of cSCC (*Nissinen et al., 2016*). In order to understand the influence of TME on the occurrence and development of cSCC, we analyzed the non-keratinocytes immune, dendritic (DC)/Mac, and stromal cells and their cell communications in cSCC samples.

Tumor-infiltrating lymphocytes are the main components of TME, especially T lymphocytes play an important role in immune response to tumor antigens. We identified 11 lymphocyte subpopulations by re-clustering, including CD4+ T cells (CD4T), regulatory T cells (Treg), naive CD4+ T cells (CD4Tnaive), naive CD8+ T cells (CD8Tnaive), CD8+ effector T cells (CD8Teff), exhausted CD8+ T cells (CD8Tex), natural killer T cells (NKT), proliferating T cells, CD4- CD8- naive T cells (DNT), B cells, and plasma cells (*Figure 7A*). In poorly differentiated cSCC sample, the CD8Tex subpopulation accounted for a considerable proportion, which were small in moderately differentiated/well-differentiated cSCC samples (*Figure 7B*). The CD8Tex cells have higher expression of inhibitory receptors, and the effector function is significantly reduced or lost, which may be one of the main factors of immune dysfunction. Ji et al. have found that this group of cells were mainly located at the edge of inflammatory response and in immune cell clusters in cSCC (*Ji et al., 2020*). Besides, proliferating T cells also had a higher percentage in poorly differentiated cSCC sample. On the contrary, the proportion of CD4Tnaive decreased with the increase of malignant degree (*Figure 7B*).

DCMac cells are important antigen-presenting cells in skin tissues, which play a key role in initiating, regulating, and maintaining immune response (*Kashem et al., 2017*). In this study, eight subpopulations were identified in cSCC samples (*Figure 7A*). Among them, the stable monocyte-derived DCs (moDCs) had characteristic high expression of MRC1 and ITGAX (*Figure 7A*). The cluster with high expression of LAMP3 and CCR7 was identified as mature myeloid DCs (mmDC). It is a mature form of conventional DC cells, which has the potential to migrate from tumor to lymph node and can interact with a variety of T lymphocytes. LC overexpressed CD1A, CD1C, and CD207. The immature conventional type I DC cells (cDC1) expressed unique C-type lectin receptor CLEC9A and chemokine receptor XCR1 (*Figure 7A*). They can cross-present antigen and promote anti-tumor immune response of CD8+ T cells. Meanwhile, type II DC cells (cDC2) expressed CLEC10A. The proliferating DC highly expressed CD11c/ITGAX and proliferating markers. The cluster with high expression of IRF4 and CLEC4C was identified as plasma DC (pDC). Another cluster highly expressed CD68 and HLA-DR, and represent Mac (*Figure 7A*).

The stromal cells include fibroblasts and endothelial cells. Cancer-associated fibroblasts (CAFs) are one of the most important members of TME, interacting with tumor cells and playing an important

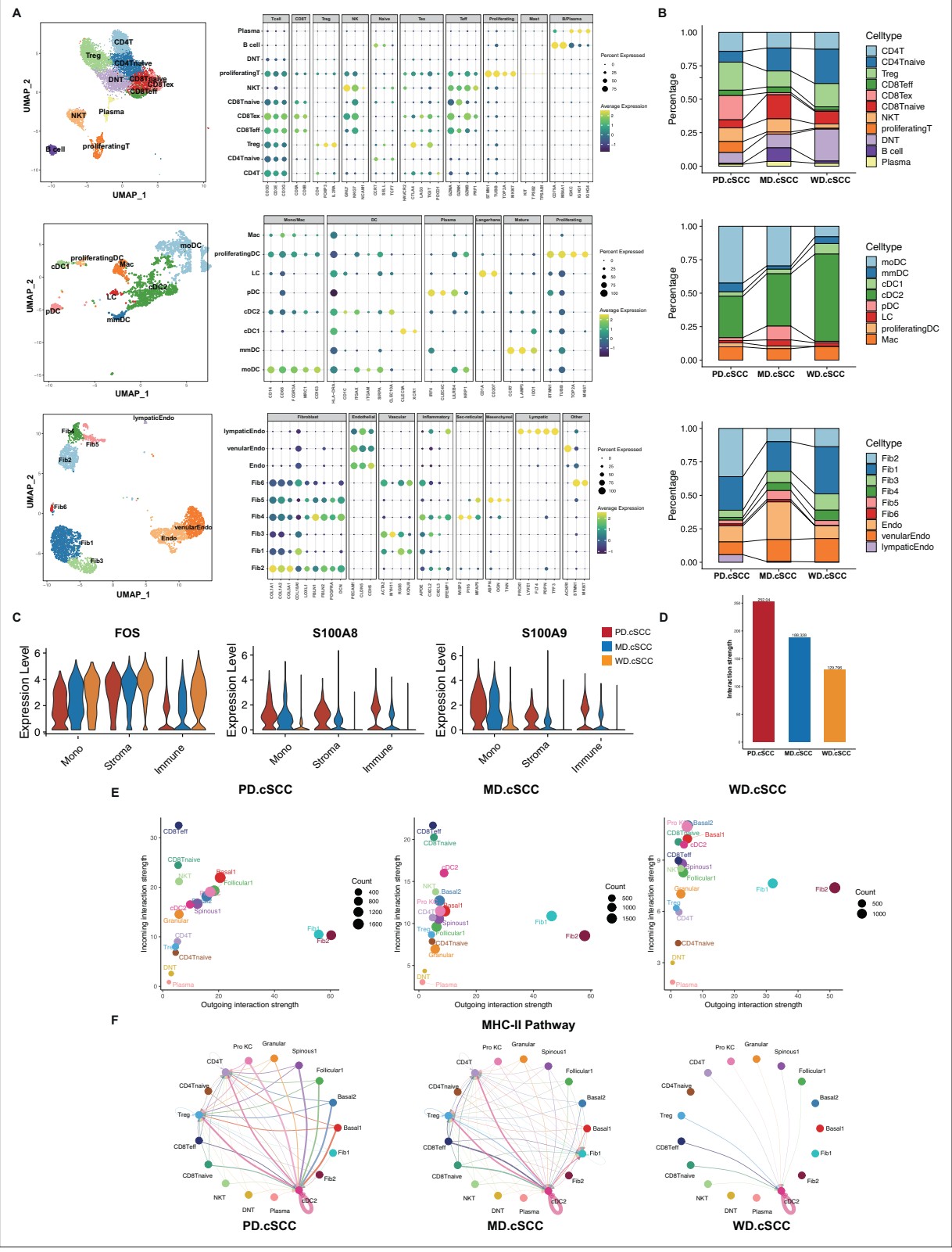

**Figure 7.** The analysis of cell-cell communication in tumor microenvironment (TME) of cutaneous squamous cell carcinoma (cSCC) samples. (**A**) Identification of TME cell subpopulations based on their marker genes, including T cells, dendritic (DC) cells, and stromal cells. (**B**) Cell proportion of cell subpopulations in T cells, DC cells, and stromal cells, respectively. (**C**) The monotonically changed differentially expressed genes (DEGs) in TME cells in cSCC samples of all stages (Wilcoxon test, p_val_adj <0.05). (**D**) Comparison of cell interactions among the different clinical stages of cSCC (Wilcoxon

*Figure 7 continued on next page*

*Figure 7 continued*

test, ****p<0.0001). (**E**) Comparison of total incoming path weights vs total outgoing path weights across all cell populations in three cSCC samples, respectively (Wilcoxon test, p<0.1). (**F**) Circle plot showing the inferred intercellular communication signaling strength network of MHC-II pathway in three cSCC samples, respectively (Wilcoxon test, p<0.1).

The online version of this article includes the following figure supplement(s) for figure 7:

**Figure supplement 1.** The second-level clustering on fibroblasts.

**Figure supplement 2.** The heatmap and functional analysis of top 15 differentially expressed genes (DEGs) in fibroblast subsets.

role in the occurrence and development of tumor (*Sahai et al., 2020*). CAFs can secrete a variety of growth factors, cytokines, and ECM proteins to promote tumor cell invasion and metastasis. In this study, we identified six subpopulations (Fib1–6) of CAFs (*Figure 7A* and *Figure 7—figure supplement 1A*). Fib1 highly expressed typical perivascular cell markers including RGS5, KCNJ8, ACTA2, and MCAM (*Figure 7A* and *Figure 7—figure supplement 1B*; *Crisan et al., 2012*; *Sweeney and Foldes, 2018*). There are also literatures suggesting that these genes are pericytes markers (*He et al., 2016*; *Kirkwood et al., 2021*). However, the typical pericytes markers CSPG4 and PDGFB are barely expressed in Fib1 (*Figure 7—figure supplement 1B*). The enrichment analysis of top markers in Fib1 showed the related function on angiogenesis (*Figure 7—figure supplement 2*). It suggested the similarity between Fib1 and perivascular fibroblasts. Fib2 were canonical fibroblasts, which expressed COL1A1/2, LUM, DCN, and other typical markers. Fib3 highly expressed ACTA2 and MYH11 (*Figure 7A*). The myofibroblasts are usually defined by ACTA2 and have ultrastructural characteristics between fibroblasts and VSMC (*Bagalad et al., 2017*). Besides, Fib3 had a high expression of AOC3, which has been reported as a specific marker distinguishing myofibroblasts from other fibroblasts (*Figure 7—figure supplement 1C*; *Hsia et al., 2016*). It suggested the similarity between Fib3 and myofibroblasts. Fib4 both highly expressed secretory reticular fibroblast markers MGST1, MFAP5, WISP2 and papillary fibroblast marker PTGDS (*Figure 7A* and *Figure 7—figure supplement 1D*; *Ascensión et al., 2021*). The functional analysis of Fib4 markers indicated that this group was related to cell adhesion and immune cell migration (*Figure 7—figure supplement 2*). The secretory reticular fibroblasts have been reported contributing to ECM organization and immunomodulatory (*Bensa et al., 2023*). Fib5 specifically expressed typical mesenchymal fibroblast markers like ASPN, OGN, and TNN (*Ascensión et al., 2021*). Fib6 expressed proliferating fibroblast markers such as STMN1, MKI67, and TOP2A (*Figure 7A*). Besides, there are three subpopulation of endothelial cells including Endo, venularEndo, and lympaticEndo (*Figure 7A*).

In TME cells, we also identified DEGs among the different groups. Several key genes showed monotonically changing trends associated with disease progression. For example, with the increase of malignancy, FOS showed down-regulation while S100A8 and S100A9 monotonically increased in all three types of TME cells (*Figure 7C*).

## Cell-cell communication analysis revealed important signaling pathways related in cSCC tumor

To investigate the effect of TME on invasive cSCC, CellChat, a cell communication analysis tool for single-cell transcriptome data, was used to analyze the intercellular interactions of cSCC samples based on gene expression data, ligand-receptor database information, and cell communication reference database (CellChatDB) (*Jin et al., 2021*). As shown in *Figure 7D*, with the increase of malignant degree, the strength of intercellular interactions showed an increasing tendency in cSCC samples (Wilcoxon test, ****p<0.0001). Furthermore, with the increase of malignant degree, the incoming interaction strength of keratinocytes including Basal1 and Basal2 in poorly differentiated/moderately differentiated cSCC tumors significantly enhanced than in well-differentiated sample (*Figure 7E*, *Supplementary file 1a7*, Wilcoxon test, p<0.1). The outgoing interaction strength of stromal cells including Fib1 and Fib2 was always strong in all stages of cSCC (*Figure 7E*). Besides, the difference of cell-cell communication in distinct stages of cSCC was also found in cancer-related signaling pathways. For example, the cell interaction through MHC-II pathway was enhanced with the increased malignancy of cSCC. In poorly differentiated/moderately differentiated cSCC samples, MHC-II pathway exhibited more abundant signaling interactions among all cell types than in well-differentiated sample (*Figure 7F*, *Supplementary file 1r*, Wilcoxon test, p<0.1).

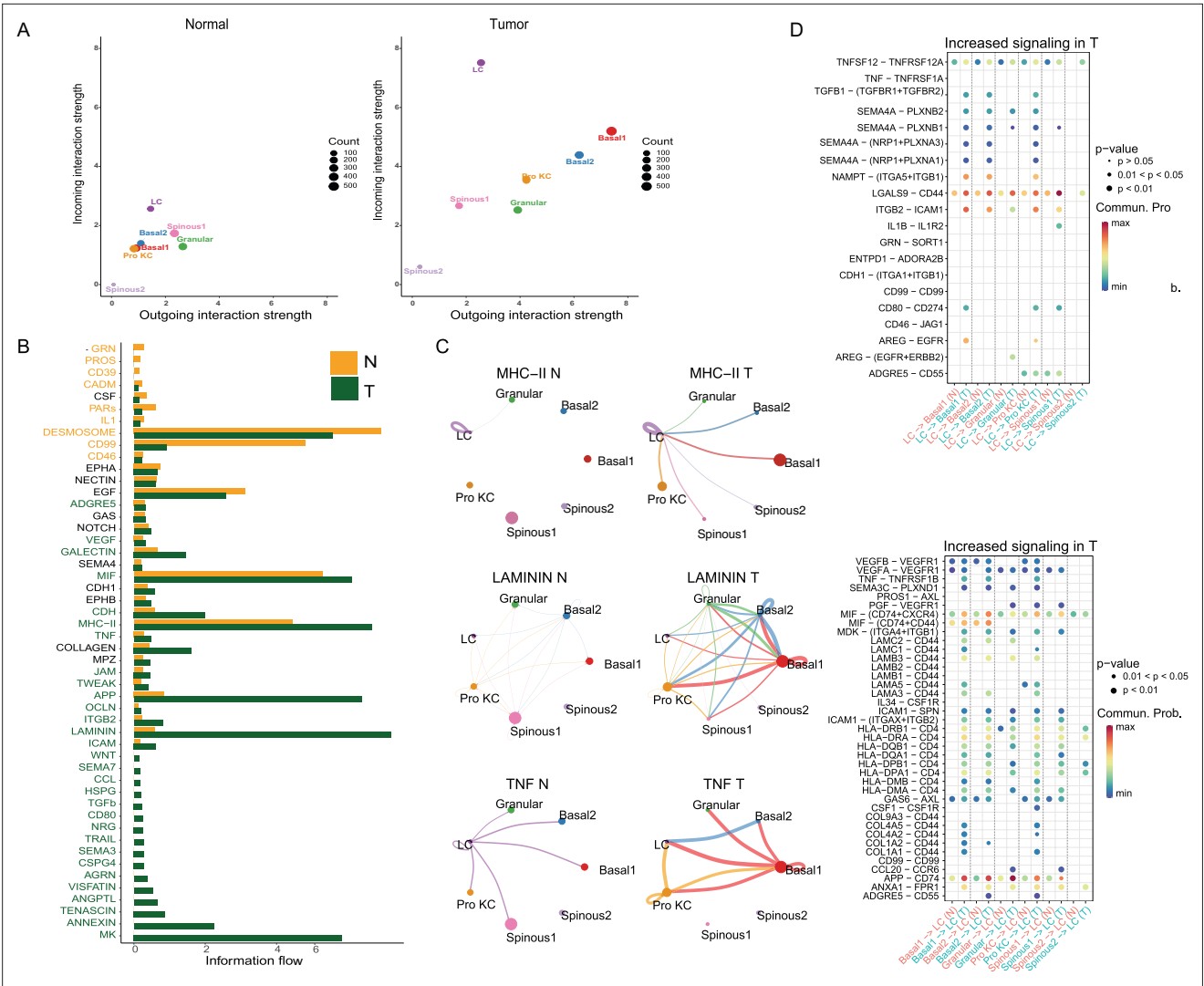

**Figure 8.** The analysis of cell-cell communication in tumor microenvironment (TME) of poorly differentiated cutaneous squamous cell carcinoma (cSCC) and matched normal samples. (**A**) Comparison of total incoming path weights vs total outgoing path weights between normal and tumor samples across common cell populations (Wilcoxon test, p<0.1). (**B**) Significant signaling pathways were ranked based on differences in the overall information flow within the inferred networks between normal and tumor samples. The signaling pathways colored orange are enriched in normal tissue, and pathways colored dark green were enriched in the tumor tissue (Wilcoxon test, p<0.1). (**C**) Circle plot showing the inferred intercellular communication signaling strength network between normal and tumor samples in MHC-II, LAMININ, and TNF pathway (Wilcoxon test, p<0.1). (**D**) Comparison of the significant ligand-receptor pairs between normal and tumor skin. The top shows the contribution to the signaling from Langerhans to KC subpopulations. The bottom shows the contribution to the signaling from KC subpopulations to Langerhans. Dot color reflects communication probabilities and dot size represents computed p-values. Empty space means the communication probability is zero. p-Values are computed from one-sided permutation test.

The online version of this article includes the following figure supplement(s) for figure 8:

**Figure supplement 1.** The analysis of cell-cell communication in tumor microenvironment (TME) of poorly differentiated cutaneous squamous cell carcinoma (cSCC) and matched normal samples.

**Figure supplement 2.** Chromatin accessibility is associated with transcription factor activity.

We also compared cell-cell communication in poorly differentiated cSCC sample with its matched normal sample. It was found that cell-to-cell interaction was significantly enhanced in tumor (**Figure 8—figure supplement 1A**, Wilcoxon test, p<0.1), and multiple interaction pathways were significantly active. Both of the incoming and outgoing interaction strength in Basal1, Basal2 and Pro KCs and LC cells were significantly increased, suggesting that these cell populations were the most important participants in cell crosstalk with TME in cSCC (**Figure 8A**, **Supplementary file 1s**, Wilcoxon test, p<0.1).

Further analysis revealed that some classical cancer-related signaling pathways were significantly altered in invasive cSCC (*Figure 8B*, *Supplementary file 1t*, Wilcoxon test, p<0.1). Basal cells can interact with various immune cells through MHC-II, laminin, and TNF signaling pathways in poorly differentiated cSCC sample (*Figure 8C and D*, *Supplementary file 1u*, Wilcoxon test, p<0.1). These enhancement of cell interactions through these three signaling pathways were also verified by CellPhoneDB approach (*Figure 8—figure supplement 1B*, Wilcoxon test, p<0.1). Studies have found that some tumor cells can play the role of antigen presentation by up-regulation of MHC-II expression on cell surface, inducing differentiation and invasion of Tregs cells and participating in tumor genesis (*Chaoul et al., 2018*). Laminin is an important component of ECM, which is involved in basement membrane skeleton formation, cell adhesion, growth, differentiation, and migration (*Cloutier et al., 2019*). Laminin signaling pathways have been proven to regulate the morphology, differentiation, and movement of a variety of cells including keratinocytes (*Yap et al., 2019*). They can also participate in signal transmission and promote tumor infiltration and metastasis (*Rousselle and Scoazec, 2020*). In cSCC, the expressions of ligand-receptors corresponding to laminin signaling pathway were significantly enhanced in keratinocytes, suggesting that laminin signaling pathway has important significance for the occurrence and development of cSCC. The TNF family and its receptors play key roles in a variety of immune and inflammatory process. It plays a dual role in tumor, not only playing an immunomodulatory and tumor suppressive role, but also promoting tumor immune escape by inducing inflammatory response, promoting tumor cell survival, proliferation, and EMT, regulating Treg and bone marrow-derived suppressor cells (*Sheng et al., 2018*). These results suggested that these signaling pathways and related cell subpopulations may play key roles in cSCC.

## Chromatin accessibility is associated with transcription factor activity

To understand the chromatin accessibility and its role on regulation of gene expression in invasive cSCC, we also performed scATAC-seq on tumor sample of poorly differentiated individual with scRNA-seq to generate paired, cell-type-specific chromatin accessibility and transcriptional profiles. We leveraged the annotated scRNA-seq dataset of tumor sample to predict scATAC-seq cell types with Seurat using label transfer. Comparison between scATAC-seq cell-type predictions and curated annotations of scRNA-seq dataset indicated that all major cell types were present in both datasets (*Figure 8—figure supplement 2A*). Then, we detected accessible chromatin regions and investigated differentially accessible chromatin regions (DARs) between cell types (*Supplementary file 1v*). The majority of DARs were located in a promoter region within 1 kb of the nearest transcriptional start site, especially in Basal1 the proportion was more than 95% (*Figure 8—figure supplement 2B*). Meanwhile, a lot of DARs were closely associated with DEGs in their respective cell types and we could distinguish cell groups in scATAC-seq dataset based on the state of DARs. For example, the ATAC peaks of KRT5 were increased in major keratinocytes. CD83 is one of the best markers for mature DCs (*Lechmann et al., 2002*). The coverage plot showed an increase in number and amplitude of ATAC peaks within its promoter and gene body in DCs and Langerhans (*Figure 8—figure supplement 2C*).

Given that many transcription factors may be key determinants in the development of cSCC in our above results, we used chromVAR to infer transcription factor-associated chromatin accessibility in scATAC-seq dataset of cSCC tumor. We observed that the cell types also could be distinguished by transcription factor activities, suggesting these cell-type-specific transcription factors could regulate chromatin accessibility. For example, as well-known driver genes in cSCC, TP63, and TP53 were detected an enrichment of their binding motifs within DAR in major keratinocytes (*Figure 8—figure supplement 2D* and *Supplementary file 1w*) that was supported by increased chromatin accessibility and increased transcription in the scRNA-seq of them. More importantly, similar patterns were seen for FOSL1, which were identified as potential key driver transcription factors for the development of cSCC in our above results (*Figure 8—figure supplement 2D* and *Supplementary file 1w*). In addition, predicted cis-regulatory chromatin interactions by Cicero also showed a positive correlation between transcription factor activity and expression on a global level (*Figure 8—figure supplement 2E*). Meanwhile, the above key transcription factors including TP63 and FOSL1 showed a positive correlation between motif activity and expression, further supporting their important driving roles in the development of cSCC as transcriptional activators (*Figure 8—figure supplement 2F* and *Supplementary file 1x*).

## Discussion

The occurrence and development of tumor is an extremely complex process. Although many studies have been carried out, the key mechanisms of the occurrence and development of AK and cSCC are still elusive. In this study, the gene expression profile information from 13 samples of six patients was obtained by single-cell transcriptomic sequencing, covering all stages of cSCC development including normal skin, AK, SCCIS, and invasive cSCC, and 138,982 high-quality single cells were finally obtained for analysis.

The six normal skin tissues near the lesions were all from the exposed sites of elderly individuals, showing obvious photoaging, but the histopathology did not show morphological abnormality of keratinocytes. Nine different main cell clusters were identified from normal skin tissues and further subgroup analysis found different subtypes in basal, spinous, and follicular cells. There have been reports about the subgroups of epidermal cells in previous studies (*Ji et al., 2020*; *Cheng et al., 2018*). In our study, according to the expression of marker genes, Basal1 subgroup had high expression of the main components of hemidesmosomes COL17A1 and stem cell marker genes, suggesting that Basal1 might be stationary Basal cells attached to the basement membrane (*Ghadially, 2012*). Although Basal2 subgroup had high expression of basal cell-related markers, the expression of COL17A1 and stem cell-related genes was decreased, suggesting they may be the basal cells that have finished division and are about to directionally differentiate. The Pro KCs highly expressed MKI67 and also had basal cell markers. They are transient amplifying cells with strong proliferating ability, which can leave from the basal layer after limited mitosis and enter the process of terminal differentiation and migration (*Eckert et al., 2013*). Follicular cells can also be divided into Follicular1 and Follicular2 subgroups. Follicular2 had higher expression of WNT pathway-related suppressor genes such as SFRP1, FRZB, and DKK3 than that of Follicular1. Cheng et al. also identified these cells in human normal skin by single-cell sequencing technology and speculated that these cells were stem cell groups in the protuberant of the outer root sheath of hair follicles (*Cheng et al., 2018*; *Lim et al., 2016*).

The up-regulation of IGFBP2 was reported in both murine and human basal cell carcinoma (BCC), which promoted BCC development by mediating epidermal progenitor cell expansion via Hedgehog (Hh) signaling pathway (*Villani et al., 2010*). In our study, transcriptional level of main targets of Hh signaling pathway was analyzed to evaluate the activity of this pathway, and results showed no significant difference across cell types between normal and AK samples (*Figure 2—figure supplement 4*). It suggested that IGFBP2 might mediate AK development through other signaling pathways, which requires further investigation.

Our results showed that the percentage of basal cells in SCCIS increased significantly compared with AK and normal samples. This was even more significant in cSCC, especially in the Basal1 subpopulation with high expression of stem cell-related markers, while Pro KCs only slightly increased. These results suggested that cell terminal differentiation may be impaired during the development of cSCC, and that basal cells play a more important role than Pro KCs.

Although the origin of cSCC has always been controversial, it is believed that basal cells with rapid proliferation ability and differentiated keratinocytes may all be the origin cells of cSCC (*Goldie et al., 2019*). However, there are other studies that suggest that stem cells such as static epidermal stem cells and hair follicle stem cells are the most important origin cells of cSCC (*Martin et al., 2016*). Morris et al. found that skin tumors came from stationary, 5-fluorouracil-insensitive epidermal stem cells rather than rapidly proliferating epidermal cells in mouse chemo-carcinogenic model of cSCC (*Morris et al., 1997*). Adriana et al. also found that epidermal stem cells were the main origin cells of BCC, and proliferating epidermal cells mainly caused benign proliferative skin lesions (*Sánchez-Danés et al., 2016*).

To identify the malignant keratinocytes in AK, SCCIS, and cSCC, we performed CNV analysis. Significant CNV differences were identified in cSCC samples, and the CNV differences among the samples were significant, which was proportional to the histopathological classification and risk grade, confirming the significant heterogeneity among the cSCC samples. However, we did not identify obvious CNV in AK samples, which may be related to the low proportion of malignant cells or the mild degree of malignancy in AK samples. In SCCIS samples, we also identified some cells with CNV differences, which may be the keratinocytes with early malignant transformation. Identification of malignant cells from SCCIS and invasive cSCC showed that these cells were mainly derived from stationary basal cells, some Pro KCs, Follicular2 cells, and a small number of spinous cells, again suggesting

the importance of basal cells. Therefore, we focused on the characterization of these cells in SCCIS to understand the key events that promote the progression of precancerous lesions or carcinoma in situ to invasive cSCC. The characteristic marker genes of basal cells of SCCIS were compared with normal and AK basal cells. Two subpopulations of basal cells in SCCIS were identified based on CNV scores and re-clustering and Basal-SCCIS-tumor with higher CNV score were identified as malignant cell group.

Ji et al. identified TSK subgroup in European cSCCs by using scRNA-seq and revealed their spatial distribution in combination with spatial transcriptomics (*Ji et al., 2020*). In this study, we did not identify identical TSK cell subpopulations, which may be related to ethnic differences, different sample sources, and significant heterogeneity between tumors. Besides, the subpopulation of TSKs in their study only accounted for 2.7–13.8% of all tumor cells. However, in our study, we found a unique subpopulation in early stage of cancer with certain invasive characteristics in SCCIS. This subpopulation highly expressed KRT15, COL17A1, MAGEA4, LAMC2, which is consistent with findings in TSK. We used SingleCellNet (*Tan and Cahan, 2019*) to compare the Basal-SCCIS-tumor/normal with the populations defined in Ji et al. The attribution plot and classification heatmap showed a strong similarity between Basal-SCCIS-tumor and Tumor_KC_Basal subcluster in Ji et al. Importantly, about 10% of Basal-SCCIS-tumor cells were classified as TSK (*Figure 4—figure supplement 4A and B*). In cSCC tumor samples, the basal cells also contained Tumor_KC_Basal and TSK populations in Ji et al. (*Figure 4—figure supplement 4C and D*). The basal tumor cells of cSCC samples in our study highly expressed CCL2, CXCL14, FTH1, MT2A, which is consistent with the findings in Tumor_KC_Basal.

We compared the expression profiles of the same cell subpopulations during progression of AK, SCCIS, and cSCC and identified a group of important candidate genes in each disease stage. ALDH3A1, IGFBP2, DYNC1H1, NFKBIZ, and RND3 were specifically highly expressed in AK. ALDH3A1 is a newly discovered tumor stem cell marker in recent years. Overexpression of ALDH3A1 in melanoma and lung cancer not only regulates tumor cell stemness and the process of EMT, but also promotes inflammation through up-regulation of inflammatory factors such as COX2 and PGE2, and enhances the expression of PD-L1 to affect immune escape (*Terzuoli et al., 2019*). In vitro studies have confirmed that IGFBP2 is involved in regulating the proliferation, invasion, and metastasis of tumor cells. IGFBP2 secreted in melanoma activates the PI3K/Akt pathway to promote tumor angiogenesis by binding to integrin αVβ3 (*Zhao et al., 2018*). However, the expression of ALDH3A1 and IGFBP2 was not elevated in the three cSCC samples, suggesting that they may play different regulatory roles in different stages of AK and cSCC.

It is of concern that two basal subgroups with different levels of CNV were identified in SCCIS, and the expression of a large number of HSPs was generally increased in the Basal-SCCIS-tumor subgroup with higher level of CNV. Fernandez et al. found that HSP70 was increased in the cytoplasm of keratinocytes in cSCC tissues arising from AK and was positively correlated with dermal infiltration level (*Fernández-Guarino et al., 2020*). It may be an early potential marker of progression from AK to cSCC. The high expression of activated keratin genes and S100 family genes also indicated the high invasiveness of Basal-SCCIS-tumor subgroup, which had great potential to transform to invasive cSCC.

Among Basal-SCCIS-tumor subgroup-specific genes, MAGEA4 was confirmed to be strongly positive in most SCCIS and invasive cSCC by IHC. MAGEA4 has been proven to inhibit p53-dependent apoptosis in cancer cells, enhance aggressivity of tumor cells, and induce cellular and humoral immune responses (*Coles et al., 2020*). Importantly, it was also identified as one of TSK markers in Ji et al.'s paper (*Ji et al., 2020*). Therefore, we inferred that MAGEA4 is a new biomarker of higher malignancy in certain SCCIS individuals, although it needs further studies. In addition, ITGA6 and other tumor-related genes were also significantly overexpressed in Basal-SCCIS-tumor. They may play an important role in the progression of SCCIS to cSCC by regulating cell stemness, cell proliferation, cytoskeleton, and ECM degradation. We also identified a group of closely related significantly up-regulated genes in cSCC. The function of genes that have received little attention in the past was validated at the cellular level. Our functional experiments found that LGALS1, IFITM3, FTH1, BST2, and SAT1 genes affected the proliferation, apoptosis, migration, and invasion of human cSCC.

In TME analysis of cSCC samples, we identified major subpopulations of immune, DC/Mac, and stromal cells based on their specific markers. In cell communication analysis, we found that the interactions of keratinocytes and TME cells were enhanced with the increase of malignant degree in cSCC

tumor samples. In cell-cell communication analysis of poorly differentiated cSCC and patient-matched normal sample, we observed significantly enhanced cell-to-cell interactions in cSCC tumor sample. The increased interactions mainly enriched in basal cells with TME cells. In addition, we identified several crucial cancer-related signaling pathways in cSCC. The activation of these signaling pathways may play important regulatory function in cSCC tumor genesis and development.

In this study, we performed comprehensive analysis of scRNA-seq profiles in diverse samples to simulate the classic carcinogenic process from photoaged skin to AK, then to SCCIS, and finally to invasive cSCC. Especially, we deeply analyzed the AK as precancerous lesions and the SCCIS at the single-cell level and identified the key malignant cell subpopulation, which is significantly important to investigate the transformation from AK to cSCC. The results are significantly benefited to understand the occurrence and development of cSCC.

## Materials and methods
### cSCC and AK patient samples
cSCCs, AKs, and patient-matched normal adjacent skin samples were collected during surgical treatment at the Dermatology Department of the First Affiliated Hospital of Kunming Medical University (Yunnan, China). All AK and cSCC samples were derived from the UV-exposed areas from immunocompetent patients, and none of these patients had received any treatment before surgery. These fresh resected biopsies were divided into two parts: half of each sample was immediately dissociated into single-cell suspension for single-cell sequencing, and another half was formalin-fixed for pathological grading and immunohistochemical studies. Written informed consent for the samples obtained under protocols was approved by the Ethics Committee of the First Affiliated Hospital of Kunming Medical University. Diagnosis of all samples was confirmed by at least two independent pathologists. Histological grades of cSCC were performed according to Broder's grading system and the risk classification was performed according to the 2019 European Association of Dermato-Oncology (EADO) guidelines. And we divided AK lesions into three categories: AK I, AK II, and AK III, based on the abnormal cells in the percentage of intraepidermal neoplasia, as proposed by *Werner et al., 2015*.

### Tissue dissociation
All fresh skin samples were gently washed in RPMI 1640 after removing crust, subcutaneous fat, and necrotic tissue with surgical scissors and cutting the tissue into small pieces of 2–4 mm in a sterile tissue-culture dish. For tumor samples digestion was performed using tumor dissociation kit for human (130-095-929, MACS Miltenyi Biotec), mechanical dissociation using gentleMACS Dissociator running the gentleMACS program h_tumor_01. In order to capture enough keratinocytes in AK samples for subsequent research, we separate the epidermal tissue from the dermis and then dissociated into single-cell suspensions by combining mechanical dissociation with enzymatic degradation of the extracellular adhesion proteins using Epidermis Dissociation Kit for Human (130-103-464, MACS Miltenyi Biotec). The dissociated cell suspension was strained with a 40 µm filter (BD Falcon), and treated with Red Blood Cell Lysis Solution (130-103-183, MACS Miltenyi Biotec) and dead cell removal using the Dead Cell Removal Kit (130-090-101, MACS Miltenyi Biotec) to confirm cell viability >85% with trypan blue staining (Invitrogen). All samples were processed as per the manufacturer's instructions. Sorted cells were centrifuged and resuspended in PBS+0.04% BSA (Gibco) to a final cell concentration of 700–1200 cells/µL as determined by hemacytometer.

### 10x scRNA-seq library preparation and sequencing
The single-cell capturing and downstream library constructions were performed using the Chromium Single Cell 3' v3 (10x Genomics) library preparation kit according to the manufacturer's protocol. Cellular suspensions were co-partitioned with barcoded gel beads to generate single-cell gel bead-in-emulsion (GEM) and polyadenylated transcripts were reverse-transcribed. Incubation of the GEMs produces barcoded, full-length cDNA from polyadenylated mRNA, and amplified via PCR to generate sufficient mass for library construction. Then, the libraries were sequenced on NovaSeq6000 (Illumina).

## Nuclei isolation, 10x scATAC-seq library construction, and sequencing

The isolation, washing, and counting of nuclei suspensions were performed according to the manufacturer's protocol (10x Genomics, CG000169). Briefly, 100,000–1,000,000 cells were centrifuged at 300×*g* for 5 min at 4°C, removed the supernatant, and 100 µL chilled lysis buffer (10 mM Tris-HCl, 10 mM NaCl, 3 mM $MgCl_2$, 0.1% Tween-20, and 1% BSA) was added and incubated for 5 min on ice. Following lysis, nuclei were resuspended in chilled Diluted Nuclei Buffer (10x Genomics; PN-2000153) at approximately 5000–7000 nuclei/µL based on the starting number of cells and immediately used to generate scATAC-seq libraries. scATAC-seq libraries were prepared according to the manufacturer's protocol of Chromium Single Cell ATAC Library Kit (10x Genomics, PN-1000087). Nuclei of cSCC cells were incubated by Tn5 transposable enzymes (10x Genomics; 2000138) for 60 min at 37°C to form DNA fragments. Then, mononuclear GEMs with special 10x barcodes were generated using a microfluidic platform (10x Genomics). Next, we collected single-cell GEMs and conducted linear amplification in a C1000 Touch Thermal cycler. Emulsions were coalesced using the Recovery Agent and cleaned up using Dynabeads. Indexed sequencing libraries were then constructed, purified, and sequenced on NovaSeq6000 (Illumina).

## scRNA-seq data processing

Reads were processed using the Cell Ranger pipeline (3.1.0) with default and recommended parameters. FASTQs generated from Illumina sequencing output were aligned to the human reference genome GRCh38-3.0.0. Next, Gene-Barcode matrices were generated for each individual sample by counting unique molecular identifiers and filtering non-cell-associated barcodes. Finally, we generated a gene-barcode matrix containing the barcoded cells and gene expression counts. This output was then imported into the Seurat (4.0.5) R toolkit for quality control and downstream analysis of our scRNA-seq data. All functions were run with default parameters, unless specified otherwise. Low-quality cells (<200 genes/cell, <3 cells/gene, and >10% mitochondrial genes) were excluded. Before incorporating a sample into our merged dataset, we individually inspected the cells-by-genes matrix of each as a Seurat object.

## scATAC-seq data processing

The chromatin accessibly analysis of scATAC-seq data referred to the pipeline by *Muto et al., 2021*. The gene activity matrix was log-normalized prior to label transfer with the aggregated scRNA-seq Seurat object using canonical correlation analysis. Differential chromatin accessibility between cell types was assessed with the Signac (1.4.0) 'FindMarkers' function. Genomic regions containing scATAC-seq peaks were annotated with ChIPSeeker (1.26.2) and clusterProfiler (3.18.1) using the UCSC database on hg38. Transcription factor activity was estimated using chromVAR (1.12.0). The positional weight matrix was obtained from the JASPAR2018 database. Cis-coaccessibility networks were predicted using Cicero (1.8.1).

## Identification of cell types and subtypes by nonlinear dimensional reduction

The Seurat package implemented in R was applied to identify major cell types. Highly variable genes were generated and used to perform principal component analysis (PCA). Significant principal components (PCs) were determined using JackStraw analysis and visualization of heatmaps focusing on PCs 1–20. PCs 1–10 were used for graph-based clustering (at res = 0.5 for samples) to identify distinct groups of cells. These groups were projected onto UMAP analysis run using previously computed PCs 1–10. We characterized the identities of cell types of these groups based on expression of known markers: basal cells (COL17A1, KRT5, KRT14), spinous cells (KRT1, KRT10), granular cells (FLG, LOR), proliferating keratinocytes (Pro KCs, MKI67, TOP2A), follicular cells (KRT6B, KRT17, SFRP1), LC (CD207, CD1A), T cells (CD3D, PTPRC), melanocytes (PMEL, TYRP1), and fibroblasts (DCN, COL1A1). Subclustering of basal cells was further performed with the same approach.

## Cluster markers identification

The cluster-specific marker genes were identified by running the FindConservedMarkers function in the Seurat package to the normalized gene expression data. The DEGs were identified by the 'find. markers' function with default parameters and filtered by p_val_adj <0.05. Just in DEG analysis of

Basal-SCCIS-tumor, we used the parameter avg_log2FC >0.58 and p_val_adj <0.05 to further narrow down the gene sets. We used Metascape (http://metascape.org) to perform biological process enrichment analysis with the DEGs in each cluster or subpopulation.

## CNV estimation

Initial CNVs for each region were estimated by inferCNV (1.6.0) R package. The CNVs of total cell types were calculated by expression level from single-cell sequencing data for each cell with -cutoff 0.1 and -noise_filter 0.1. In order to well study the CNV level in keratinocytes for each tumor sample, we used the keratinocytes from patient-matched normal adjacent skin as background.

## H&E, IHC, and IF staining

For hematoxylin and eosin (H&E), formalin-fixed, paraffin-embedded cSCCs, AKs, and patient-matched normal adjacent skin biopsies were cut at 4 μm and stained using (H&E). IHC and IF staining was performed using DAB or DAPI and the following primary antibodies: anti-ALDH3A1 mouse monoclonal antibody (Santa Cruz Biotechnology), anti-BST2 rabbit polyclonal antibody (Proteintech), anti-FTH1 rabbit polyclonal antibody (Zen-Bioscience), anti-LGALS1 rabbit polyclonal antibody (Zen-Bioscience), anti-MAGEA4 rabbit monoclonal antibody (Cell Signaling Technology), anti-IFITM3 rabbit polyclonal antibody (Zen-Bioscience), anti-IGFBP2 rabbit monoclonal antibody (Abcam), anti-ITGA6 rabbit polyclonal antibody (Zen-Bioscience), anti-SAT1 rabbit polyclonal antibody (Bioss). Examination and photographic documentation were performed using a digital slide scanner-PANNORAMIC 1000 (3DHISTECH, Hungary). Histological sections were analyzed semiquantitatively. The staining intensity of IHC section was scored as 0 (negative), 1 (weak), 2 (medium), or 3 (strong). Extent of staining was scored as 0 (<5%), 1 (5–25%), 2 (26–50%), 3 (51–75%), and 4 (>75 %) according to the percentages of the positive staining areas in relation to the whole carcinoma area. Scores for staining intensity and percentage positivity of cells were then multiplied to generate the immunoreactivity score (IRS) for each case. The IF staining was analyzed with Image-ProPlus software 6.0. Integrated option density (IOD) of interesting area (AOI) was measured and density mean (IOD/AOI) were calculated as the semiquantitative parameters.

## Cell culture, transfections

Human immortalized epidermal keratinocytes cell line (HaCaT cells) and human cSCC cell line A431, SCL-I, and SCL-II (*Boukamp et al., 1982*; *Tilgen et al., 1983*) were used in this study and were obtained from the American Type Culture Collection (ATCC) and Free University of Berlin. Mycoplasma detection was carried out in cell lines using GMyc-PCR Mycoplasma Test Kit (YEASEN, 40601ES20) to avoid mycoplasma contamination. The cells were cultured in Dulbecco's modified Eagle's medium (DMEM, Gibco, USA) supplemented with 10% fetal bovine serum (FBS, Gibco) and 1% penicillin-streptomycin (Gibco) in a 5% $CO_2$ incubator at 37°C. siRNA transfections were carried out using transfections reagent (INVI DNA RNA Transfections Reagent, Invigentech, USA) to inquire into the influence of silencing gene expression on cell growth, proliferation, invasion, and metastasis. The sequences of various siRNA oligonucleotides used in this study were listed in *Supplementary file 1y*. The transfection efficiency was confirmed by RT-PCR.

## Quantitative real-time PCR

To verify the expression and siRNA transfection efficiency of key genes in cSCC cells, the mRNA expression levels of genes in A431, SCL-I, and SCL-II cells were detected by quantitative real-time PCR (qRT-PCR). Primers were designed and listed in *Supplementary file 1z*. Total RNA was extracted from cells using TRIzol reagent (Invitrogen, Thermo Fisher Scientific, USA) and reverse-transcribed into cDNA using a FastKing-RT Reagent kit (Tiangen, Beijing, China) according to the manufacturer's protocols. qRT-PCR was performed using SYBR Green Master Mix (Tiangen, Beijing, China). The RNA expression level of target genes was evaluated by $2^{-\Delta\Delta Ct}$.

## CCK8 assay

Cell-counting kit 8 (CCK8) assay was employed for the evaluation of cell proliferation. The transfected cells were seeded in a 96-well plate at a seeding density of 2000 cells/well (100 μL). Next,10 μL CCK8 reagent (Beyotime, Shanghai, China) was added to each well and incubated at 37°C. Cell proliferation

rate was assessed according to the optical density value (450 nm) detected by Microplate reader (BioTek, USA) at 0 hr, 24 hr, 48 hr, and 72 hr following the manufacturer's instructions.

## Annexin V and PI staining detects cell apoptosis

The apoptosis rate was evaluated using the Annexin V-FITC and PI kit (Beyotime, Shanghai, China) following the manufacturer's protocols. A431, SCL-I, and SCL-II cells were transfected and suspended, and 5 µL Annexin V-FITC and 10 µL PI were added to the cell suspension. After 20 min of incubation at room temperature in the dark, the cells were analyzed by FCM (FACS Cabibur; BD, CA, USA).

## Cell migration assay

A wound-healing assay was performed to test the migration ability of the cSCC cells after transfection. Cells were grown to confluence in six-well plates, and the wounds were made in confluent monolayer cells using a sterilized 200 µL pipette tips. The culture medium was then removed, and the cells were washed with PBS and cultured with the indicated treatment. Wound healing of different groups was detected at 0 hr, 24 hr, 48 hr, and 72 hr within the scraped lines, and representative fields were photographed at the different time points to assess the migratory ability of the cells.

## Transwell invasion assays

The invasion ability of cSCC cells was evaluated by Transwell assays using Transwell chambers (8 µm pore size, Corning Costar, USA) precoated with Matrigel. After transfection, $4 \times 10^4$ cSCC cells in the 200 µL serum-free medium were added to the upper chambers, DMEM with 10% FBS was added to the bottom chambers. After incubation at 37°C for 36 hr, the cells invaded into the lower side of the inserts were fixed in 4% paraformaldehyde and stained with 0.1% crystal violet. Then counted and photographed under a microscope.

## TME analysis

On the basis of the general cell population, we extracted non-keratinocyte immune cells, DC/Mac cells, and stromal cells individually for further subdivision. The extracted subsets were reintegrated using 'harmony' and then dimension reduction and clustering were performed. For subgroup cell clustering, cells of different types were extracted separately and clustered by their respective parameters (T cells: 21 PCs, using resolution of 0.5 [CD8_T: 18 PCs, using resolution of 0.7; CD4_T: 20 PCs, using resolution of 1]; DC: 24 PCs, using resolution of 0.9; Fib: 26 PCs, using resolution of 0.3). The annotations of cell identity on each subcluster were defined by the expression of known marker genes. Immune cells: Treg (CD4, FOXP3, IL2RA); CD4Tnaive (CD4, CCR7); CD8Teff (HAVCR2, CTLA4, LAG3, TIGIT); CD8Tnaive (CCR7, TCF7); NKT (CD3, NKG7, GNLY); proliferatingT (MKI67, TOP2A, TUBB, STMN1); DNT (CD3, CD4-, CD8-); B cell (CD79A, MS4A1); plasma (Ig). DC/Mac cells: moDC (MRC1, CD11c/ITGAX, cDC1 (CLEC9A, XCR1); cDC2 (CD1C, CLEC10A); pDC (IRF4, CLEC4C); LC (CD1C, CD1A, CD207); mmDC (LAMP3, CCR7); proliferating DC (CD11c/ITGAX, MKI67, STMN1); Mac (CD68, HLA-DR, CD1C-, CD11c-). Stromal cells: venularEndo (ACKR1); lympaticEndo (PROX1, LYVE1, FLT4) (*Solé-Boldo et al., 2020*).

Inference of intercellular communications was conducted using 'CellChat'. The comparison of cell communication strength among different samples included subclusters with more than 50 cells in each sample. In subsequent pathway comparisons, we retained the common subclusters in order to visually compare changes in cell communication at different stages. The ligand-receptor interactions were all based on 'CellChatDB', a database of literature-supported ligand-receptor interactions in both mouse and human. The majority of ligand-receptor interactions in CellChatDB were manually curated on the basis of KEGG signaling pathway database. The identification of major signals for specific cell groups and global communication patterns was based on an unsupervised learning method, non-negative matrix factorization.

To test the significance of enhanced cell-to-cell interaction, we obtained the communication strength using CellChat, and combined the communication strength according to cell-type pairs, where the communication strength of missing values was set to 0. Then, we conducted a paired Wilcoxon test to obtain the corresponding p-value to detect whether the enhancement of cell-to-cell interaction between two groups was significant. For the comparison of outgoing interaction strength of the same cell types between two groups, we first extracted the communication strength of each

signal pathway contributing to outgoing strength, and then merged the two sets of strength based on the signal pathway, where the communication strength of the missing signal pathway was determined to be 0. Subsequently, we performed a paired Wilcoxon test to define the significance. The comparison for incoming interaction strength is the same as for outgoing interaction strength. We defined p-value <0.1 as significance.

## Statistical analysis

All experiments were performed in triplicate technical replicates, and all data are presented as mean ± standard deviation (SD). Differences among groups were analyzed using Student's t test or one-way analysis of variance (ANOVA) for normally distributed data and the Kruskal-Wallis test for non-normally distributed data. And $p<0.05$ was considered to be statistically significant.

## Acknowledgements

The authors acknowledge the editors and reviewers for their positive and constructive comments and suggestions related to this study. Funding. This work was supported by Yunnan Science and Technology Leading Talents Project (2017HA010), Yunnan Province Clinical Research Center for Skin Immune Diseases (2019ZF012), Yunnan Province Clinical Center for Skin Immune Diseases (ZX2019-03-02), Shenzhen Science and Technology Program (JCYJ20190807160011600 and JCYJ20210324124808023), China Postdoctoral Science Foundation (2020M683073), Shenzhen Outbound Postdoctoral Research Project (SZBH202001), Guangzhou Science Technology Project (201904010007), Guangdong Provincial Key Laboratory of Digestive Cancer Research (2021B1212040006), National Natural Science Foundation of China (81872299 and 82260517), and Yunnan Provincial Health Commisssion High-level Talents Scientific Research Project (2023-KHRCBZ-B13).

---

## Additional information

### Funding

| Funder | Grant reference number | Author |
|---|---|---|
| Yunnan Science and Technology Leading Talents Project | 2017HA010 | Li He |
| Yunnan Province Clinical Research Center for Skin Immune Diseases | 2019ZF012 | Li He |
| Yunnan Province Clinical Center for Skin Immune Diseases | ZX2019-03-02 | Li He |
| Shenzhen Science and Technology Program | JCYJ20190807160011600 | Xin Li |
| Shenzhen Science and Technology Program | JCYJ20210324124808023 | Xin Li |
| China Postdoctoral Science Foundation | 2020M683073 | Ya-Zhou Sun |
| Guangzhou Science Technology Project | 201904010007 | Xin Li |
| Guangdong Provincial Key Laboratory of Digestive Cancer Research | 2021B1212040006 | Xin Li |
| National Natural Science Foundation of China | 81872299 | Xin Li |
| National Natural Science Foundation of China | 82260517 | Li He |

---

| Funder | Grant reference number | Author |
| --- | --- | --- |
| Yunnan Provincial Health Commisssion (High-level Talents Scientific Research Project) | 2023-KHRCBZ-B13 | Dan-Dan Zou |
| Shenzhen Outbound Postdoctoral Research Project | SZBH202001 | Ya-Zhou Sun |

The funders had no role in study design, data collection and interpretation, or the decision to submit the work for publication.

## Author contributions

Dan-Dan Zou, Resources, Data curation, Formal analysis, Supervision, Validation, Investigation, Visualization, Writing – original draft, Project administration, Writing – review and editing; Ya-Zhou Sun, Data curation, Formal analysis, Supervision, Funding acquisition, Validation, Investigation, Visualization, Writing – original draft, Project administration, Writing – review and editing; Xin-Jie Li, Data curation, Formal analysis, Investigation, Visualization, Writing – original draft, Writing – review and editing; Wen-Juan Wu, Resources, Formal analysis, Investigation, Project administration, Writing – review and editing; Dan Xu, Resources, Formal analysis, Validation, Investigation, Writing – review and editing; Yu-Tong He, Resources, Data curation, Visualization; Jue Qi, Ying Tu, Yang Tang, Yun-Hua Tu, Resources, Validation, Investigation; Xiao-Li Wang, Validation, Investigation; Xing Li, Feng-Yan Lu, Ling Huang, Heng Long, Resources, Investigation; Li He, Conceptualization, Resources, Data curation, Supervision, Funding acquisition, Investigation, Project administration, Writing – review and editing; Xin Li, Conceptualization, Data curation, Supervision, Funding acquisition, Investigation, Project administration, Writing – review and editing

## Author ORCIDs

Dan-Dan Zou http://orcid.org/0000-0002-0527-3599
Li He http://orcid.org/0000-0002-3601-3036
Xin Li http://orcid.org/0000-0001-8328-4894

## Ethics

Human subjects: The authors are accountable for all aspects of the work in ensuring that questions related to the accuracy or integrity of any part of the work are appropriately investigated and resolved. All procedures performed in this study involving human participants were in accordance with the Declaration of Helsinki (as revised in 2013). This study protocol was approved by the Ethics Committee of the First Affiliated Hospital of Kunming Medical University (Approval Number (2020)-L-29), and written informed consent was obtained from all patients.

## Decision letter and Author response

Decision letter https://doi.org/10.7554/eLife.85270.sa1
Author response https://doi.org/10.7554/eLife.85270.sa2

# Additional files

## Supplementary files

• Supplementary file 1. The tables of supplementary data. (a) Clinical characteristic of patients and samples enrolled in single-cell sequencing. (b) The gene list of up-regulated differentially expressed genes (DEGs) in actinic keratosis (AK) Basal1 subpopulation. (c) The gene list of up-regulated DEGs in AK Basal2 subpopulation. (d) The gene list of up-regulated DEGs in AK Pro KC subpopulation. (e) AK candidate driver genes and antibodies for IF. (f) The gene list of overlapped up-regulated DEGs in Basal subpopulation of P2 from AK vs normal and squamous cell carcinoma in situ (SCCIS) vs AK. (g) The gene list of overlapped down-regulated DEGs in basal subpopulation of P2 from AK vs normal and SCCIS vs AK. (h) The gene list of up-regulated DEGs between Basal-SCCIS-tumor vs Basal-SCCIS-normal. (i) SCCIS candidate driver genes and antibodies for immunohistochemistry (IHC). (j) The gene list of up-regulated DEGs in cutaneous squamous cell carcinoma (cSCC) Basal1 subpopulation. (k) The gene list of up-regulated DEGs in cSCC Basal2 subpopulation. Supplementary file (l) The gene list of up-regulated DEGs in cSCC Pro KC subpopulation. (m)

The gene list of up-regulated DEGs in cSCC Follicular2 subpopulation. (n) The gene list of up-regulated DEGs in cSCC Spinous1 subpopulation. (o) The gene list of up-regulated DEGs in cSCC Spinous2 subpopulation. (p) cSCC candidate driver genes and antibodies for IHC. (q) Significance test for interaction strength of cells in *Figure 7E*. (r) Significance test for interaction strength of MHC-II pathway in *Figure 7F*. (s) Significance test for interaction strength of cells in *Figure 8A*. (t) Significance test for signaling pathways in *Figure 8B*. (u) Significance test for the interaction strength of three signaling pathways of cells in *Figure 8C*. (x) The correlation between chromVAR transcription factor activity with expression in single-cell ATAC sequencing (scATAC-seq) data. (y) The sequences of various small interfering RNA (siRNA) oligonucleotides used in this study. (z) The primers of genes used for quantitative real-time PCR (qRT-PCR). (v) The differentially accessible chromatin regions between cell types of poorly differentiated cSCC sample in scATAC-seq data. (w) The chromVAR transcription factor activity between cell types of poorly differentiated cSCC sample in scATAC-seq data.

• MDAR checklist

## Data availability

The raw data and gene counts table are available from GEO under accession number (GSE193304). All data needed to evaluate the conclusions in the paper are present in the paper and/or the Supplementary Materials.

The following dataset was generated:

| Author(s) | Year | Dataset title | Dataset URL | Database and Identifier |
|---|---|---|---|---|
| He Y, Sun Y, Zou D, Li X, He Li | 2023 | Single-cell Sequencing Highlights Heterogeneity and Malignant Progression in Actinic Keratosis and Cutaneous Squamous Cell Carcinoma | https://www.ncbi.nlm.nih.gov/geo/query/acc.cgi?acc=GSE193304 | NCBI Gene Expression Omnibus, GSE193304 |

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
