## [Editor Report]

This important study delineates the molecular changes driving the progression from actinic keratosis (AK) to cutaneous squamous cell carcinoma (cSCC). Using state-of-the-art single-cell RNA profiling of 138,982 cells from 13 samples of six patients including AK, squamous cell carcinoma in situ (SCCIS), cSCC, and their matched normal tissues, thus covering comprehensive clinical courses of cSCC, the authors provide an invaluable data resource. This study identified several previously unreported and interesting candidate genes involved in different stages of the malignant progression of skin neoplasias, which have been validated in situ, and partially in vitro. These findings substantially advance our understanding of the molecular changes leading to skin cancer.

---

## [Decision Letter]

**Decision letter after peer review:**

Thank you for submitting your article "Single-cell Sequencing Highlights Heterogeneity and Malignant Progression in Actinic Keratosis and Cutaneous Squamous Cell Carcinoma" for consideration by *eLife*. Your article has been reviewed by 2 peer reviewers, one of whom is a member of our Board of Reviewing Editors, and the evaluation has been overseen by Tony Ng as the Senior Editor. The reviewers have opted to remain anonymous.

Essential revisions:

1) Analysis of the scRNAseq data needs improvement. Especially the analysis of the TME was done superficially. Second level clustering of distinct cell types and subsets within the tumor stroma should be done and DEG lists/heatmaps should be provided, and differences in these subsets with disease progression should be assessed. Further, cell proportion analysis has to be revised to allow specific claims.

2) In addition, a comparison to other published data sets should be done to strengthen the claims. E.g., presence of basal tumor cells in the Ji et al. Cell 2020 dataset.

3) Cell-cell communication analysis was performed only in 1 sample and not validated by another method. A comparison to other tumor stages would be appreciated.

4) The methodology lacks a lot of detail throughout the paper. Please, clarify how data analysis was done, the rationale and selection criteria for the presented candidate genes and markers and cell subsets, which statistical tests were performed, and how significance is defined.

5) Some of the claims regarding in vitro and in situ validation of candidate genes including MAGEA4, ITGA6, IGFBP, and ALDH3A1 are not supported by the data. If these data cannot be provided, revise the text accordingly. In addition, clarify whether there is a correlation between gene expression with the clinical progression and cSCC differentiation status (AK versus SCCIS versus poorly and well-differentiated cSCC). Xenograft assays with cSCC cell lines with MAGEA2 or ITGA6 knock-down would be important to prove the in vivo relevance of these genes in cSCC progression.

6) Please, revise the manuscript text so that extensive data discussion is not found within the Results section but in the Discussion section in order to make it easier to follow the results. Further, highlight and streamline the key findings.

*Reviewer #1 (Recommendations for the authors):*

Figure 2 E, F: The authors show increased expression of ALDH3A1 in epidermal cells of AK skin compared to normal skin. What is the expression level in SCCIS and cSCC? Does ALDH3A1 overexpression in keratinocytes affect UV-induced DNA damage response? Is ALDH3A1 expression induced by UV light in vitro?

Further, in line with increased expression of IGFBP2 in basal cells of AK samples – does IGFBP2 overexpression affect epidermal cell proliferation or invasion in vitro? Since IGFBP2 is also increased in BCC, is Hedgehog signaling activated in IGFBP2+ cells of AK?

Figure 4: in lines 353-356 the authors write "we further screened out the candidate genes that were not only highly expressed in SCCIS samples but also highly expressed in important keratinocytes in cSCC tumor samples, which play an important role in the progression of SCCIS to invasive cSCC". – Please define "important keratinocytes"! What characterizes these cells? enriched CNVs?

Figure S2D: the resolution of the IF images is too low to see co-localization of the 3 proteins. Furthermore, it would help if the border of the epithelial cells to the stroma is marked.

Figure 4K/ lines 375-376: "This inferred that MAGEA4 might become a promising new biomarker and target for the different subtypes of SCCIS with different invasive state." In Figure 4K no discrimination between poorly and well-differentiated cSCCs is shown. Is there a correlation between MAGEA4 expression and cSCC differentiation status?

Likewise, is there a correlation between ITGA6 expression and cSCC differentiation status?

Figure S3: The authors show that both MAGEA4 and ITGA6 affect proliferation, migration, invasiveness, and apoptosis rates of immortalized keratinocytes and cSCC cell lines. Xenograft assays with cSCC cell lines with MAGEA2 or ITGA6 knock-down would be important to prove the in vivo relevance of these genes in cSCC progression.

Figure 5A-C/ lines 408-410: "There are more basal cells in tumor samples than in normal skin, which indicated the loss of terminal differentiation in tumor basal cells". Is this the case for each individual pair of patient samples?

Figure 7: This analysis has been done on 1 patient sample only (poorly differentiated cSCC). The analysis highlights some interesting features, but it would have been great to see the differences or commonalities of the TME among AK and cSCC (or even different cSCC stages).

What are the DEGs that define the 3 CAF subsets?

*Reviewer #2 (Recommendations for the authors):*

Line 201. CDKN2A expression is too low to reach any meaningful conclusion.

Lines 240 – 295, all these findings are just based on one sample of SCCIS. All these need to be validated in a larger cohort.

Lines 244 – 248, rewrite, difficult to understand.

Line 298, an increased proportion of basal cells in SCCIS was only based on one sample. Could this be validated in publicly available scRNA-seq datasets?

Line 371 – line 376. MAGEA4 is full of speculations. Was MAGEA4 exclusively expressed by basal cells only? What was the rationale for investigating this gene? Is MAGEA4 expression associated with the clinical staging of the tumour? None of these were shown.

Line 399.- 401, the claim of MAGEA4 and ITGA6 regulating cell stemness, apoptosis, and ECM degradation was not supported by the data in the paragraph. Just speculation

Line 439. CDKN2A (p16) loss/deletion is one of the common genomics events in AK and cSCC. Can authors comment on why the expression of CDKN2A is up-regulated in cSCC in most of the keratinocyte cell types?

[Editors’ note: further revisions were suggested prior to acceptance, as described below.]

Thank you for resubmitting your work entitled "Single-cell Sequencing Highlights Heterogeneity and Malignant Progression in Actinic Keratosis and Cutaneous Squamous Cell Carcinoma" for further consideration by *eLife*. Your revised article has been evaluated by Tony Ng (Senior Editor) and a Reviewing Editor.

The manuscript has been improved but there are some remaining issues that need to be addressed, as outlined below:

The analysis of the TME in Figure 7 has improved but the analysis of fibroblasts needs revision. It would be more accurate to remove the endothelial cells from the UMAP and perform second-level clustering on fibroblasts only. Moreover, the signature genes provided for the different subsets are not sufficient – e.g. RGS5 and KCNJ8 positive cells could also represent pericytes. Likewise, ACTA2 and MYH11 could mark vascular smooth muscle cells (vSMC), which sometimes cluster together with fibroblasts. In general, it should be clarified why the subsets were named "mesenchymal", "canonical" or "secretoryreticular" etc. Can you provide references for the subsets or describe the reason for the label, which suggest a certain function? A Heatmap with more DEGs would be useful. Maybe it is better to label them as Fib1-6 and describe similarities to previously published subsets, if available.

Furthermore, in Figure 4—figure supplement 2 (former Figure S2D) the authors have not marked the border of epithelium and stroma but rather separated the tumor tissue from adjacent skin. A demarkation of the basement membrane would help to discriminate cancer cells from the stroma.

---

## [Author Response]

Essential revisions:1) Analysis of the scRNAseq data needs improvement. Especially the analysis of the TME was done superficially. Second level clustering of distinct cell types and subsets within the tumor stroma should be done and DEG lists/heatmaps should be provided, and differences in these subsets with disease progression should be assessed. Further, cell proportion analysis has to be revised to allow specific claims.

Thank you for your valuable suggestions. We have provided the second-level clustering of distinct cell types and subsets within the tumor stroma, DEGs and differences in these subsets with disease progression in new Figure 7. The descriptions of results were added in ‘The tumor micro-environment (TME) landscape of cSCC’ section in main text.

We improved the cell proportion analysis by providing the significance test of all cell types in distinct clinical groups. These results were presented in new Figure 2—figure supplement 1 and Figure 5—figure supplement 1.

2) In addition, a comparison to other published data sets should be done to strengthen the claims. E.g., presence of basal tumor cells in the Ji et al. Cell 2020 dataset.

We compared basal tumor cell in our study with the cell populations defined in Ji et al. Cell 2020 dataset using SingleCellNet [1]. The results showed that both the Basal-SCCIS-tumor cells of SCCIS and basal tumor cells of cSCC in our study closely resemble the Tumor_KC_Basal subcluster defined in Ji et al’s paper (Figure 4—figure supplement 4, C and D). Tumor_KC_Basal highly expressed CCL2, CXCL14, FTH1, MT2A, which is consistent with our findings in basal tumor cells.

3) Cell-cell communication analysis was performed only in 1 sample and not validated by another method. A comparison to other tumor stages would be appreciated.

We added TME analysis for all cSCC samples and presented these results in ‘The tumor micro-environment (TME) landscape of cSCC’ section and new Figure 7. We also used another approach cellphoneDB as suggested to verify our cell-cell communication results. There are 55-58% of the ligand-receptor interactions predicted by CellChat were also predicted by CellPhoneDB (Figure 3 in this letter). The enhancement of cell interaction through MHC-II, Laminin and TNF signaling pathways in poorly-differentiated cSCC sample compare to normal sample were consistent in both CellChat and CellPhoneDB (Figure 8C and Figure 8—figure supplement 1B).

4) The methodology lacks a lot of detail throughout the paper. Please, clarify how data analysis was done, the rationale and selection criteria for the presented candidate genes and markers and cell subsets, which statistical tests were performed, and how significance is defined.

The first consideration of candidate gene selection is the fold change of expression. Then we selected top changed genes and searched a large amount of literature for these genes. Combined with functional enrichment analysis, we gave priority to those genes that were not been reported direct relation to cSCC development but have close relationship with related pathways. These genes were arranged for further verification experiments. We have added more details of selection criteria in main text and methods section.

5) Some of the claims regarding in vitro and in situ validation of candidate genes including MAGEA4, ITGA6, IGFBP, and ALDH3A1 are not supported by the data. If these data cannot be provided, revise the text accordingly. In addition, clarify whether there is a correlation between gene expression with the clinical progression and cSCC differentiation status (AK versus SCCIS versus poorly and well-differentiated cSCC). Xenograft assays with cSCC cell lines with MAGEA2 or ITGA6 knock-down would be important to prove the in vivo relevance of these genes in cSCC progression.

We have added more experiment to validate the function of MAGEA4, ITGA6, IGFBP, and ALDH3A1. More results were also provided to investigate the correlation between these gene expression with the clinical progression and cSCC differentiation status. The detailed descriptions can be found below and in corresponding part in main text and figures.

We have tried to conduct xenograft assays with A431 cells. However, due to the small tumor volume and the low tumor formation rate, we failed to proceed to the next step of the experiment. We provided the details of experimental process and results below.

6) Please, revise the manuscript text so that extensive data discussion is not found within the Results section but in the Discussion section in order to make it easier to follow the results. Further, highlight and streamline the key findings.

We have revised all the paper carefully and adjusted the content in article to make it easier to read.

Reviewer #1 (Recommendations for the authors):Figure 2 E, F: The authors show increased expression of ALDH3A1 in epidermal cells of AK skin compared to normal skin. What is the expression level in SCCIS and cSCC? Does ALDH3A1 overexpression in keratinocytes affect UV-induced DNA damage response? Is ALDH3A1 expression induced by UV light in vitro?Further, in line with increased expression of IGFBP2 in basal cells of AK samples – does IGFBP2 overexpression affect epidermal cell proliferation or invasion in vitro? Since IGFBP2 is also increased in BCC, is Hedgehog signaling activated in IGFBP2+ cells of AK?

The upregulation of ALDH3A1 expression was found to be specific to AK samples with no significant upregulation observed in SCCIS and cSCC, indicating the unique role of ALDH3A1 in precancerous lesions of skin. We have added these results in Figure 2—figure supplement 3, A and B and main text.

To explore the relationship between ALDH3A1 expression and UV irradiation, HaCat cells were subjected to UVB irradiation followed by analysis of ALDH3A1 expression. The results showed that ALDH3A1 expression decreased in a dosage-dependent manner upon UVB irradiation (Figure 2—figure supplement 3C) in contrast to the increased ALDH3A1 level observed in AK samples. The discrepancy between in vivo and in vitro results may be attributed to the shorter duration of UVB treatment in vitro. Moreover, the development of AK is a long dynamic process, during which many comprehensive alterations other than short-time UVB irradiation may upregulate ALDH3A1 expression [2]. These results were presented in Figure 2—figure supplement 3C, and the discrepancy was discussed in the main text.

To investigate the effects of IGFBP2 on epidermal cells, we performed overexpression of IGFBP2 in HaCaT and A431 cells and measured the cell proliferation and cell invasion. The results showed that the IGFBP2 overexpression significantly promoted the proliferation and invasiveness of both HaCaT cells and A431 cells (*p* < 0.05, Figure 2—figure supplement 3, D and E).

Transcriptional level of main target of Hedgehog (Hh) signaling pathway were analyzed to evaluate the activity of this pathway, and results showed no significant difference across cell types between normal and AK samples (Figure 2—figure supplement 4). It suggests that IGFBP2 might mediate AK development through other signaling pathways, which requires further investigation. We have discussed these results in the main text.

Figure 4: in lines 353-356 the authors write "we further screened out the candidate genes that were not only highly expressed in SCCIS samples but also highly expressed in important keratinocytes in cSCC tumor samples, which play an important role in the progression of SCCIS to invasive cSCC". – Please define "important keratinocytes"! What characterizes these cells? enriched CNVs?

Thanks for this suggestion. Here the “important keratinocytes” means the keratinocytes with proliferative capacity or differentiative potential, including Basal1, Basal2, Pro KC and Follicular1. We have revised the texts in manuscripts accordingly.

Figure S2D: the resolution of the IF images is too low to see co-localization of the 3 proteins. Furthermore, it would help if the border of the epithelial cells to the stroma is marked.

Thanks for this suggestion. We have replaced with the high-resolution IF images and provided the enlarged image with marked cell border in Figure 4—figure supplement 2.

Figure 4K/ lines 375-376: "This inferred that MAGEA4 might become a promising new biomarker and target for the different subtypes of SCCIS with different invasive state." In Figure 4K no discrimination between poorly and well-differentiated cSCCs is shown. Is there a correlation between MAGEA4 expression and cSCC differentiation status?Likewise, is there a correlation between ITGA6 expression and cSCC differentiation status?

Thanks for the suggestion. We analyzed the expression levels of MAGEA4 and ITGA6 in cSCC of different stages. Results showed that the expression of MAGEA4 was not significantly different among SCCIS, WD cSCC and MD/PD cSCC, suggesting MAGEA4 might be activated continuously in SCCIS and cSCC tumors of all stages. These new results were presented in Figure 4—figure supplement 1. Besides, MAGEA4 is identified as one of TSK markers in Ji et al.’s paper [3]. Taken together, we inferred that MAGEA4 is a new biomarker of higher malignancy in certain SCCIS individuals, although it needs further studies.

ITGA6 expression was also analyzed. Results showed that invasive cSCC had a higher level of ITGA6 compared with SCCIS, while there was no significant difference between the WD cSCC and MD/PD cSCC. These results were presented in new Figure 4—figure supplement 1D.

Figure S3: The authors show that both MAGEA4 and ITGA6 affect proliferation, migration, invasiveness, and apoptosis rates of immortalized keratinocytes and cSCC cell lines. Xenograft assays with cSCC cell lines with MAGEA2 or ITGA6 knock-down would be important to prove the in vivo relevance of these genes in cSCC progression.

Thank you very much for this valuable suggestion. We agree that xenograft assays will help clarify the in vivo relevance of genes identified in our study. We conducted the xenograft assays with A431 cells and athymic BALB/c-nu mice, but the general take rate was low, around 25-30% (5 out of 20, 3 out of 10). Moreover, the volume of tumor (1/2 the width^2^ X the height) was quite small (~50 mm^3^). We subsequently performed xenograft assays with sh-MAGEA4 A431 cells (sh-MAGEA4 lentivirus infected) and control cells (mock lentivirus infected). Unfortunately, as shown in Author response image 1, we were unable to observe any tumor growth with the lentivirus infected A431 cells, neither mock virus or sh-MAGEA4 virus infected cells. It is plausible that the lentivirus infection significantly reduced the tumorigenicity of A431.

Although we were unable to carry out the xenograft assay successfully, the high expression of MAGEA4 in SCCIS and cSCC samples as well as the functional validation experiments in cSCC cell line indicated its important roles. Additionally, as previously mentioned, MAGEA4 is identified a marker of tumor specific keratinocytes in Ji et al.’s paper [3]. Taken together, we inferred that MAGEA4 has close relevance to cSCC progression.

**Author response image 1. sa2fig1:** The xenograft assay results. 4-6 weeks old male BALB/c-nu mice weighing 18-22g were selected for xenograft assays. A431 cells on log phase were digested and washed twice by centrifugation with serum-free culture medium. (A) In the first xenograft assay, cell suspension concentration was 2.5*10^7^ cells/mL. Under the sterile conditions, 0.2 ml cell suspension was injected subcutaneously into the right armpit of each BALB/c-Nu mouse. After 2 weeks, 3 out of 10 mice (30%) developed subcutaneous tumors, with a volume of about 5-15 mm^3^. There was no significant increase of tumor volume after 30-days continued observation. (B) In the second xenograft assay, cell suspension concentration was adjusted to 5*10^7^ cells/mL. Matrigel matrix were added to improve the rate of tumor formation. After 2 weeks, 5 out of 20 mice (25%) developed subcutaneous tumors, and the tumor volume was about 10-80 mm^3^. There was no significant increase of tumor volume after 2-months continued observation. (C) The xenograft assays with sh-Control A431 cells (mock lentivirus infected). There were no visible tumors in sh-Control group (20 mice) after 2-months continued observation. (D) The xenograft assays with sh-MAGEA4 A431 cells (sh-MAGEA4 lentivirus infected). There were no visible tumors in sh-MAGEA4 group (20 mice) after 2-months continued observation.

Figure 5A-C/ lines 408-410: "There are more basal cells in tumor samples than in normal skin, which indicated the loss of terminal differentiation in tumor basal cells". Is this the case for each individual pair of patient samples?

Thanks for raising such a good point. Indeed, the proportion of Basal1 cells was significantly increased in tumor compared to normal skin for each patient. To visualize the increase in tumors of each patient more directly, we replotted the proportion of Basal1 cells for each individual and presented the new graphs in Figure 5—figure supplement 1.

Figure 7: This analysis has been done on 1 patient sample only (poorly differentiated cSCC). The analysis highlights some interesting features, but it would have been great to see the differences or commonalities of the TME among AK and cSCC (or even different cSCC stages).What are the DEGs that define the 3 CAF subsets?

Thank you for this suggestion. We added TME analysis for all cSCC samples and presented these results in new Figure 7. However, due to the way of sample collection, we were unable to perform TME analysis in AK samples: in order to capture enough keratinocytes in AK samples for subsequent analysis, we separate the epidermal tissue from the dermis. Because of this, there were too few TME cells in AK samples for TME analysis. We plan to do this analysis with newly collected AK samples containing complete skin tissues in the future.

Due to increase of sample numbers, the CAF of all the 3 samples were pooled and recategorized into 6 subsets in the revised manuscript. The DEGs that define these subsets are as follows: canonicalFib (COL1A1/2, FBLN1/2, PDGFRA, DCN), perivascularFib (RGS5, KCNJ8), myoFib (ACTA2, MYH11), secretoryreticularFib (WISP2, MFAP5), mesenchymalFib (ASPN, OFN) and proliferatingFib (MKI67, STMN1). The new results were updated in Figure 7A. The procedure of recategorizing CAFs was described in methods section.

Reviewer #2 (Recommendations for the authors):Line 201. CDKN2A expression is too low to reach any meaningful conclusion.

Yes, the expression level of CDKN2A is relatively low, but the expression differences between AK and normal skin in basal cells are significant. We have added the statistical data in main text (avg_log2FC = 0.49 in Basal1, avg_log2FC = 0.36 in Basal2, p_val_adj < 0.05). The violin plot of CDKN2A in Figure 2—figure supplement 2D was consistent with these data. More data of CDKN2A expression can also be found in Supplementary file 1b, 1c.

Lines 240 – 295, all these findings are just based on one sample of SCCIS. All these need to be validated in a larger cohort.

Here all the described genes have already been reported to have a close relationship with cSCC development, including SCCIS. Therefore, these genes are not our primary focus. Rather, they serve to confirm the key roles of the monotonically increased genes. Additionally, it is challenging to collect AK and SCCIS samples from the same patients, which limits our ability to validate their monotonically increased trend in more samples at this stage.

Lines 244 – 248, rewrite, difficult to understand.

We have rewritten the text.

Line 298, an increased proportion of basal cells in SCCIS was only based on one sample. Could this be validated in publicly available scRNA-seq datasets?

Unfortunately, there are currently no publicly available scRNA-seq datasets for SCCIS, which preventing us from validating the increased proportion of basal cells in SCCIS based on other datasets. However, it is worth noting that an increase of basal cell proportion has also been observed in cSCC tumors, in both our current study and other publicly available scRNA-seq datasets of cSCC [3], suggesting that increased proportion of basal cells might be a common feature of squamous skin cancers.

Line 371 – line 376. MAGEA4 is full of speculations. Was MAGEA4 exclusively expressed by basal cells only? What was the rationale for investigating this gene? Is MAGEA4 expression associated with the clinical staging of the tumour? None of these were shown.

MAGEA4 was not exclusively expressed by basal cells only (Figure 4K), but it showed significantly higher expression in Basal-SCCIS-tumor compared to Basal-SCCIS-normal (Figure 4J). We analyzed the expression levels of MAGEA4 in cSCC of different stages. Results showed that the expression of MAGEA4 was not significantly different among SCCIS, WD cSCC and MD/PD cSCC, suggesting MAGEA4 be activated continuously in SCCIS and cSCC tumors of all stages. These new results were presented in Figure 4—figure supplement 1, C and D.

MAGEA4 was investigated not only due to its higher expression level in SCCIS and cSCC, but also because of its significant role in other cancer types. MAGEA4 has been proven to inhibit p53-dependent apoptosis in cancer cells, enhance aggressivity of tumor cells, and induce cellular and humoral immune responses [13]. Furthermore, MAGEA4 was found to be highly expressed in melanoma, pancreatic cancer, lung cancer and esophageal squamous cell carcinoma [14]. Importantly, it was also identified as one of TSK markers in Ji et al.’s paper (Figure 2C in their paper), indicating its close relationship to invasive cSCC. Therefore, we inferred MAGEA4 might be a new biomarker of higher malignancy in certain SCCIS individuals, although it needs further studies.

Line 399.- 401, the claim of MAGEA4 and ITGA6 regulating cell stemness, apoptosis, and ECM degradation was not supported by the data in the paragraph. Just speculation

We apologize for the lack of figure reference, which may have caused you to miss important information. However, we would like to clarify that we performed all the necessary functional experiments to validate the role of MAGEA4 and ITGA6 in regulating cell stemness, proliferation, apoptosis, and ECM degradation, and we presented the results in Figure 4—figure supplement 3.

Line 439. CDKN2A (p16) loss/deletion is one of the common genomics events in AK and cSCC. Can authors comment on why the expression of CDKN2A is up-regulated in cSCC in most of the keratinocyte cell types?

Thanks for raising this point. There is inconsistency on CDKN2A expression level in AK and cSCC in previous reports. Although CDKN2A (p16) loss/deletion was frequently reported in AK and cSCC, there are also researches showing upregulation of CDKN2A in AK and cSCC [15, 16].

References

1 Tan Y, Cahan P. SingleCellNet: A Computational Tool to Classify Single Cell RNA-Seq Data Across Platforms and Across Species. *Cell Syst* 2019; 9: 207-213.e202.

2 Qu Y, He Y, Yang Y, Li S, An W, Li Z *et al.* ALDH3A1 acts as a prognostic biomarker and inhibits the epithelial mesenchymal transition of oral squamous cell carcinoma through IL-6/STAT3 signaling pathway. *J Cancer* 2020; 11: 2621-2631.

3 Ji AL, Rubin AJ, Thrane K, Jiang S, Reynolds DL, Meyers RM *et al.* Multimodal Analysis of Composition and Spatial Architecture in Human Squamous Cell Carcinoma. *Cell* 2020; 182: 497-514 e422.

4 Matsumura H, Mohri Y, Binh NT, Morinaga H, Fukuda M, Ito M *et al.* Hair follicle aging is driven by transepidermal elimination of stem cells via COL17A1 proteolysis. *Science* 2016; 351: aad4395.

5 Liu N, Matsumura H, Kato T, Ichinose S, Takada A, Namiki T *et al.* Stem cell competition orchestrates skin homeostasis and ageing. *Nature* 2019; 568: 344-350.

6 Nanba D, Toki F, Asakawa K, Matsumura H, Shiraishi K, Sayama K *et al.* EGFR-mediated epidermal stem cell motility drives skin regeneration through COL17A1 proteolysis. *J Cell Biol* 2021; 220.

7 Busslinger GA, Weusten BLA, Bogte A, Begthel H, Brosens LAA, Clevers H. Human gastrointestinal epithelia of the esophagus, stomach, and duodenum resolved at single-cell resolution. *Cell Rep* 2021; 34: 108819.

8 An L, Ling P, Cui J, Wang J, Zhu X, Liu J *et al.* ROCK inhibitor Y-27632 maintains the propagation and characteristics of hair follicle stem cells. *Am J Transl Res* 2018; 10: 3689-3700.

9 Ekman AK, Bivik Eding C, Rundquist I, Enerbäck C. IL-17 and IL-22 Promote Keratinocyte Stemness in the Germinative Compartment in Psoriasis. *J Invest Dermatol* 2019; 139: 1564-1573.e1568.

10 Sanz Ressel BL, Massone AR, Barbeito CG. Expression of the epidermal stem cell marker p63/CK5 in cutaneous papillomas and cutaneous squamous cell carcinomas of dogs. *Res Vet Sci* 2021; 135: 366-370.

11 Song Y, Wang B, Li H, Hu X, Lin X, Hu X *et al.* Low temperature culture enhances ameloblastic differentiation of human keratinocyte stem cells. *J Mol Histol* 2019; 50: 417-425.

12 Purdie KJ, Harwood CA, Gulati A, Chaplin T, Lambert SR, Cerio R *et al.* Single nucleotide polymorphism array analysis defines a specific genetic fingerprint for well-differentiated cutaneous SCCs. *J Invest Dermatol* 2009; 129: 1562-1568.

13 Coles CH, McMurran C, Lloyd A, Hock M, Hibbert L, Raman MCC *et al.* T cell receptor interactions with human leukocyte antigen govern indirect peptide selectivity for the cancer testis antigen MAGE-A4. *J Biol Chem* 2020; 295: 11486-11494.

14 Tang WW, Liu ZH, Yang TX, Wang HJ, Cao XF. Upregulation of MAGEA4 correlates with poor prognosis in patients with early stage of esophageal squamous cell carcinoma. *Onco Targets Ther* 2016; 9: 4289-4293.

15 Inman GJ, Wang J, Nagano A, Alexandrov LB, Purdie KJ, Taylor RG *et al.* The genomic landscape of cutaneous SCC reveals drivers and a novel azathioprine associated mutational signature. *Nat Commun* 2018; 9: 3667.

16 Yan G, Li L, Zhu S, Wu Y, Liu Y, Zhu L *et al.* Single-cell transcriptomic analysis reveals the critical molecular pattern of UV-induced cutaneous squamous cell carcinoma. *Cell Death Dis* 2021; 13: 23.

[Editors’ note: what follows is the authors’ response to the second round of review.]

The manuscript has been improved but there are some remaining issues that need to be addressed, as outlined below:The analysis of the TME in Figure 7 has improved but the analysis of fibroblasts needs revision. It would be more accurate to remove the endothelial cells from the UMAP and perform second-level clustering on fibroblasts only. Moreover, the signature genes provided for the different subsets are not sufficient – e.g. RGS5 and KCNJ8 positive cells could also represent pericytes. Likewise, ACTA2 and MYH11 could mark vascular smooth muscle cells (vSMC), which sometimes cluster together with fibroblasts. In general, it should be clarified why the subsets were named "mesenchymal", "canonical" or "secretoryreticular" etc. Can you provide references for the subsets or describe the reason for the label, which suggest a certain function? A Heatmap with more DEGs would be useful. Maybe it is better to label them as Fib1-6 and describe similarities to previously published subsets, if available.

Thank you for your valuable suggestions. We have removed the endothelial cells and performed second-level clustering on fibroblasts only. As shown in new Figure 7—figure supplement 1A, the subpopulations of fibroblasts were consistent with our previous results. However, as you mentioned, it is difficult to define the type of subpopulations due to the non-specificity of many signature genes. Thus, we labeled the fibroblasts as Fib1-6 according to your suggestions and described similarities to previously published subsets in main text. Moreover, we provided the expression of more signature genes, a heatmap and functional analysis of top15 DEGs for the different subsets in Figure 7—figure supplement 1 and Figure 7—figure supplement 2 respectively.

Furthermore, in Figure 4—figure supplement 2 (former Figure S2D) the authors have not marked the border of epithelium and stroma but rather separated the tumor tissue from adjacent skin. A demarkation of the basement membrane would help to discriminate cancer cells from the stroma.

We have added the demarcation of the basement membrane in the new figure.